# The lactonase BxdA mediates metabolic specialisation of maize root bacteria to benzoxazinoids

Lisa Thoenen [1,2], Marco Kreuzer [3], Christine Pestalozzi [2], Matilde Florean[4], Pierre Mateo [1], Tobias Züst[1,5], Anlun Wei [2], Caitlin Giroud[2], Liza Rouyer [6], Valentin Gfeller [1], Matheus D. Notter[7], Eva Knoch[6,8], Siegfried Hapfelmeier [7], Claude Becker[6,8], Niklas Schandry [6,8], Christelle A. M. Robert [1], Tobias G. Köllner [4], Rémy Bruggmann[3], Matthias Erb[1] ✉ & Klaus Schlaeppi [1,2] ✉

Root exudates contain specialised metabolites that shape the plant's root microbiome. How host-specific microbes cope with these bioactive compounds, and how this ability affects root microbiomes, remains largely unknown. We investigated how maize root bacteria metabolise benzoxazinoids, the main specialised metabolites of maize. Diverse and abundant bacteria metabolised the major compound in the maize rhizosphere MBOA (6-methoxybenzoxazolin-2(3H)-one) and formed AMPO (2-amino-7-methoxyphenoxazin-3-one). AMPO forming bacteria were enriched in the rhizosphere of benzoxazinoid-producing maize and could use MBOA as carbon source. We identified a gene cluster associated with AMPO formation in microbacteria. The first gene in this cluster, *bxdA* encodes a lactonase that converts MBOA to AMPO in vitro. A deletion mutant of the homologous *bxdA* genes in the genus *Sphingobium*, did not form AMPO nor was it able to use MBOA as a carbon source. BxdA was identified in different genera of maize root bacteria. Here we show that plant-specialised metabolites select for metabolisation-competent root bacteria. *BxdA* represents a benzoxazinoid metabolisation gene whose carriers successfully colonize the maize rhizosphere and thereby shape the plant's chemical environmental footprint.

Plant microbiomes fulfil key functions for plant and ecosystem health. Root-associated microbes promote plant growth, provide nutrients, and protect plants from pathogens[1,2]. While some root microbes are ubiquitous, many microbes form specific relationships with their host plants, and host plants often exert substantial control over the structure and function of their microbiome. Plants primarily shape their root-associated microbiome through the secretion of root exudates, which can account for up to one-fifth of the plant's assimilated carbon[3]. Root exudates may attract, nourish, or repel soil microbes and contain primary metabolites including sugars, amino acids, organic acids and fatty

---

[1]Institute of Plant Sciences, University of Bern, Bern, Switzerland. [2]Department of Environmental Sciences, University of Basel, Basel, Switzerland. [3]Interfaculty Bioinformatics Unit, University of Bern, Bern, Switzerland. [4]Department of Natural Product Biosynthesis, Max Planck Institute for Chemical Ecology, Jena, Germany. [5]Department of Systematic and Evolutionary Botany, University of Zurich, Zurich, Switzerland. [6]LMU Biocenter, Faculty of Biology, Ludwig-Maximilians-University Munich, Martinsried, Germany. [7]Institute for Infectious Diseases, University of Bern, Bern, Switzerland. [8]Gregor Mendel Institute of Molecular Plant Biology GmbH, Austrian Academy of Sciences, Vienna BioCenter (VBC), Vienna, Austria. ✉e-mail: matthias.erb@unibe.ch; klaus.schlaeppi@unibas.ch

acids, as well as secondary metabolites. The latter, also specialised metabolites, govern the plant's interactions with the environment, and among other functions, they increase biotic and abiotic stress tolerance[4]. A key function of exuded specialised metabolites is to shape the root microbiomes[5–7], documented with examples including glucosinolates, camalexins, triterpenes, and coumarins from *Arabidopsis thaliana*[5], the saponin tomatine from tomato[8], and benzoxazinoids[9–13], diterpenoids[14], zealexins[15] and flavonoids[16] from maize.

Benzoxazinoids are multifunctional indole-derived metabolites produced by *Poaceae*, including crops such as wheat, maize, and rye[17]. These compounds accumulate in leaves as chemical defences against insect pests and pathogens[17] and are exuded from the roots as phytosiderophores[18] and antimicrobials[19–21]. Benzoxazinoids directly shape the root and rhizosphere microbiomes[9–11,22], and when metabolised to aminophenoxazinones by soil microbes, they also become allelopathic, inhibiting the germination and growth of neighbouring plants[17]. In maize, the methoxylated benzoxazinoids dominate, while rye only produces non-methoxylated benzoxazinoids, and wheat forms a mixture of both[23,24]. In Supplementary Fig. 1 we document the full names, structures, and relatedness of all compounds relevant to this study. DIMBOA-Glc is the main root-exuded benzoxazinoid of maize[11], and its chemical fate in the soil is well understood. Upon exudation, plant- or microbe-derived glucosidases[17] cleave off the glucose moiety to form DIMBOA, which spontaneously converts to more stable MBOA[25]. In soil, MBOA has a half-life of several days and can be further metabolised to reactive aminophenols by microbes[17]. Three routes to different metabolite classes are known: route (I), favoured under aerobic conditions[26], forms aminophenoxazinones such as AMPO and AAMPO; route (II) results in acetamides such as HMPAA through acetylation[27], or alternatively, route (III) yields malonic acids such as HMPMA through acylation[27]. Route I is certainly relevant for the rhizosphere as the AMPO can be detected in soils of cereal fields over several months[25]. While the chemical pathways of benzoxazinoid metabolisation are well-defined, the responsible microbes and enzymes remain largely unknown.

Benzoxazinoids and their metabolisation products have antimicrobial properties. Yet, it remains poorly understood, how microbes cope with these bioactive plant metabolites[19–21,28,29]. We discriminate metabolite-microbe interactions as 'native' or 'non-host', the latter referring to context where root microbes and root metabolites do not originate from the same host. Recently, we demonstrated that 'native' root bacteria (isolated from maize) tolerated the maize-originating benzoxazinoids better compared to 'non-host' bacteria isolated from Arabidopsis[21]. Increased tolerance could either involve reduced sensitivity of molecular targets in the bacteria or metabolisation to less toxic compounds or complete degradation. Metabolisation of plant-derived compounds may not only reduce toxicity but also have added benefits for bacterial growth. For example, *Pseudomonas* isolated from Arabidopsis uses triterpenes as carbon sources[30]. *Sphingobium* bacteria isolated from tomato grow on tomatine as a sole carbon source[8]. These examples suggest that native bacteria have evolved specialised adaptations to metabolise specialised metabolites in root exudates of their host – this hypothesis remains untested.

For benzoxazinoids, the question whether native bacteria, i.e. isolated from roots or rhizosphere of benzoxazinoid-producing plants, can specifically metabolise the specialised metabolites of their host, has not been tested. There is evidence that several soil microbes can metabolise benzoxazinoids. Examples of compound conversions include APO formation from BOA (non-methoxylated form of MBOA, Supplementary Fig. 1) by *Acinetobacter* bacteria[31], formation of the acetamide HPAA from BOA by the fungus *Fusarium sambucus*[27], or accumulation of APO from BOA upon co-culture of *Fusarium verticillioides* with a *Bacillus* bacterium[32]. Testing different soil microbes from various environments revealed that they differed strongly in their metabolic activities but that degradation resulted in the expected

sequence of compounds from DI(M)BOA-Glc to DI(M)BOA to (M)BOA[13]. First insights into the molecular mechanisms include the identification of the metal-dependent hydrolase CbaA from *Pigmentiphaga* bacteria that degrade modified benzoxazinoids[33], and of a metallo-β-lactamase (MBL1) from the maize seed endophytic fungus *Fusarium verticilloides* that degrades BOA to the malonic acid HPMA[34]. Benzoxazinoid metabolisation by microbes has commonly been studied with diverse microbes isolated from different soil environments. Microbial metabolisation of benzoxazinoids and its genetic basis have not yet been investigated in the native context of root microbes from benzoxazinoid-exuding plants.

To uncover the biochemistry and the genetic basis of microbial benzoxazinoid metabolisation, we systematically screened a recently established strain collection of native maize bacteria (MRB)[21] for benzoxazinoid metabolisation. Using metabolite analyses, genetics, comparative genomics, combined with biochemical and genetic validation, we have characterised benzoxazinoid-metabolising maize root bacteria and identified the underlying genetic mechanism of AMPO formation. We identified a conserved gene cluster containing a lactonase and experimentally showed that it catalyses the degradation of MBOA. Homology searches of the key gene *bxdA* in a wide range of bacteria and plating of root microbiomes from different plant species revealed that the capacity to form AMPO is limited to maize root microbes. Our work demonstrates that maize root bacteria are adapted to metabolise the specialised metabolites of their host.

## Results

### AMPO formation is characteristic for maize root bacteria

Screening the maize root bacteria (MRB) collection for tolerance to MBOA[21], we had observed that some liquid cultures turned red, including *Sphingobium* LSP13 and *Microbacterium* LMB2 (Supplementary Fig. 2). Analysis of the liquid media by UPLC-MS revealed that these bacteria degraded MBOA and formed AMPO, which has a dark red colour. AMPO was confirmed analytically by NMR analysis. The colour change to red was also found on MBOA-containing agar plates and thus, could serve as visual phenotype to study AMPO formation. We applied this plating assay to investigate abundance and specificity of AMPO formation in root microbiomes. First, we plated microbial extracts of maize roots on control plates and plates supplemented with MBOA (Fig. 1a) and determined the proportions of red colonies. In extracts of wild-type maize we quantified ~7.7% of the root bacteria, ~5.8% of the rhizosphere bacteria and ~11.4% of the soil bacteria forming AMPO (Fig. 1b). These proportions of AMPO-forming bacteria decreased by more than 50% in extracts of the benzoxazinoid-deficient mutant *bx1*. In comparison with maize we tested root extracts of wheat (*Triticum aestivum*), which accumulates less and predominantly non-methoxylated benzoxazinoids (i.e., BOA instead of MBOA)[35–37], lucerne (*Medicago sativa*), oilseed rape (*Brassica napus*) and Arabidopsis; the latter three species do not produce benzoxazinoids. We found the highest proportion of AMPO-forming colonies on maize roots (~7.7%), followed by *Brassica* (~1%), *Triticum* (~0.5%), *Medicago* (~0.07%) and Arabidopsis (~0.002%; Fig. 1c). Together these findings indicate that AMPO formation is abundant in root microbiomes of maize and enriched by the presence of benzoxazinoids.

### Several abundant maize root bacterial genera form AMPO

To test how widespread AMPO formation is among maize root bacteria, we screened the MRB collection[21] by plating 110 strains on MBOA-containing agar plates. Based on the colony and surrounding media colour after 10 days incubation time, we classified the strains as non (no colour change compared to DMSO control plates), weak (light red colouration) or strong AMPO-formers (dark red colour; see Supplementary Fig. 3). We identified 43/110 strains, belonging to six genera from two phyla, with colour changes to light or dark red (Fig. 1d). Strong AMPO-formers were among the genera *Microbacterium* (17/

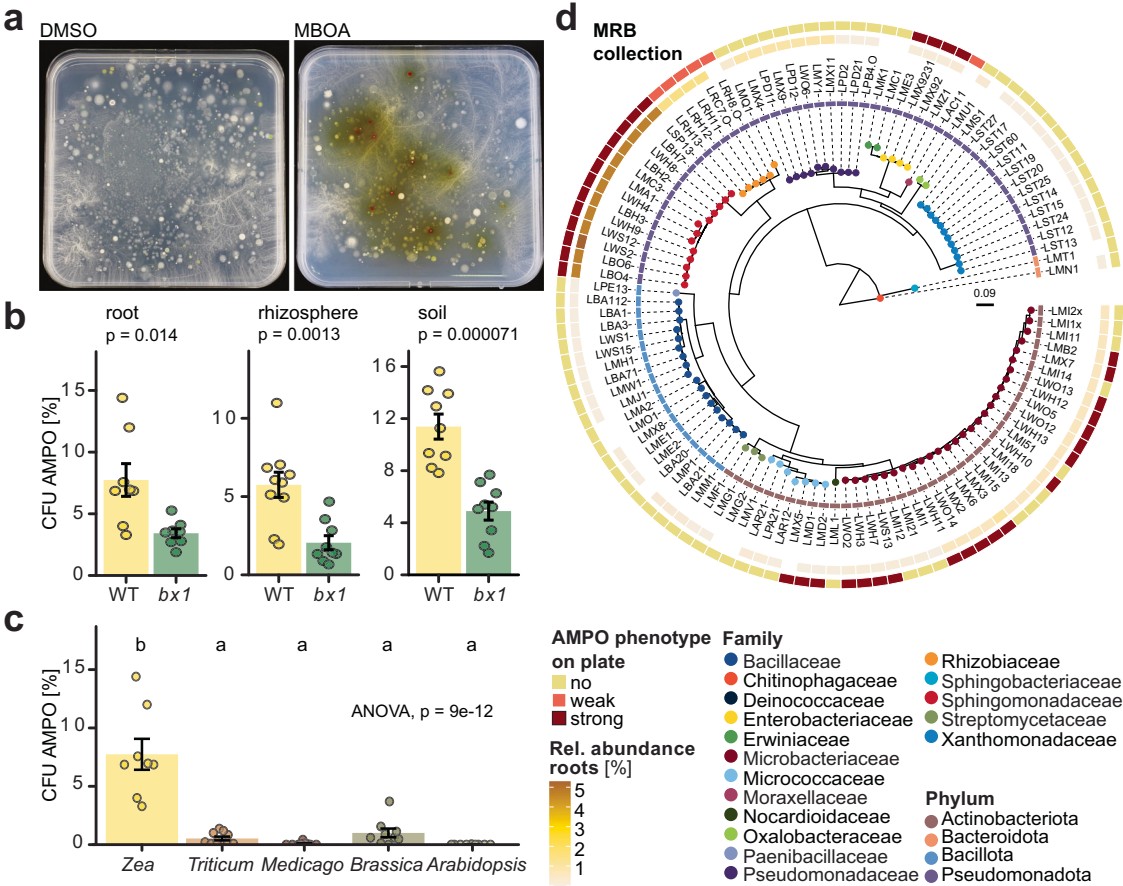

**Fig. 1 | AMPO-forming bacteria are abundant on benzoxazinoid-exuding maize roots. a–c** AMPO-forming colonies in extracts from different plant species. Root, rhizosphere or soil extracts were plated on 10% TSA supplemented with DMSO or MBOA (200 µg/mL) and cycloheximide to suppress fungal growth. Colonies were scored for AMPO formation after 10 days of incubation. **a** AMPO-forming colonies in maize root extracts. AMPO-forming colonies appear red on the MBOA-supplemented medium. **b** Percentage of total colony-forming units (CFU) that form AMPO on wild-type maize (WT) or benzoxazinoid-deficient *bx1* mutant roots, in the rhizosphere and soil. Means ± standard error and individual data points are shown (WT n = 8, *bx1* n = 9). Results of two-tailed *t*-tests are shown inside the panels. **c** Percentage of AMPO-forming CFUs in root extracts of benzoxazinoid-producing *Zea mays* (maize), *Triticum aestivum* (wheat) and non-benzoxazinoid-producing *Medicago sativa* (lucerne), *Brassica napus* (oilseed rape) and *Arabidopsis*

*thaliana*. Means ± SE and individual data points are shown (n = 10, except maize n = 8, see b) ANOVA and compact letter display of all pairwise comparisons (Significance-level: FDR-corrected p < 0.05) of estimated marginal means are shown. **d** Phylogenetic tree of the MRB strain collection. The inner ring is coloured according to the relative abundance (%) of the corresponding partial 16S rRNA gene sequence in the microbiome profile of maize roots, from which most of the isolates were obtained. The outer ring displays the phenotype of the strains on 100% TSA plates supplemented with MBOA (200 µg/mL). Strains were classified as strong AMPO-former based on a dark red colouring, weak AMPO-former for strains with a light red colour change or non-AMPO-former for strains not showing a colour change compared to the control. Tree tips are coloured by family taxonomy and the ring next to the strain IDs reports phylum taxonomy. The maximum likelihood phylogeny is based on partial 16S rRNA gene sequences.

28 strains tested) and *Pseudoarthrobacter* (3/3), *Sphingobium* (13/13) and *Enterobacter* (4/4). Weak AMPO-formers belonged to *Rhizobium* (5/6) and *Acinetobacter* (1/1). This visual identification of AMPO formation was confirmed by metabolite profiling in liquid cultures with MBOA (see below). Overall, we concluded that several genera among cultured maize root bacteria have the trait of AMPO formation.

To approximate the abundance of these cultured AMPO-forming bacteria in cultivation-independent data, we mapped the identified maize strains to published maize root microbiota datasets. First, potential AMPO-formers accounted for ~8% of the bacterial communities of the roots from which the MRB strains were isolated from[11], with *Sphingobium* contributing the most (5.3%, strong AMPO-former), followed by *Rhizobium* (1.5%, weak), *Microbacterium* (1.1%, strong) and *Enterobacter* (0.15%, strong; Fig. 1d). Second, in maize root microbiota data from fields[9], community abundances ranged from 2.9% (Changins, CH), to 7.5% (Aurora, US) and 12.0% (Reckenholz, CH; Supplementary Fig. 4). Despite phenotypic heterogeneity in AMPO formation among closely related strains (e.g., *Microbacterium*, Fig. 1d), this mapping analysis is consistent with the conclusion from cultivation-

dependent analysis (Fig. 1b) that AMPO formation is abundant in root microbiomes of maize.

## Strong MBOA-degraders are strong AMPO-formers
To chemically validate AMPO formation and to test whether maize root bacteria degrade MBOA without forming AMPO, we exposed a taxonomically broad set of 46 strains of the MRB collection in liquid cultures and quantified MBOA metabolisation using UPLC-MS. Again, we classified AMPO formation and MBOA degradation as strong, weak or no phenotypes (see "Methods" section and Supplementary Fig. 5a for cut-offs). To focus on AMPO formation and because the assay is semi-quantitative, we considered only bacteria with no or strong phenotypes. We identified 6 strains as strong MBOA-degraders (>90% degraded) and 8 strains as strong AMPO-formers, while most strains did not markedly degrade MBOA (30/46 strains) nor form AMPO (32/46; Fig. 2a). For some AMPO-formers also AAMPO was detected, which is a metabolisation product of AMPO. Of note, *Enterobacter* LME3 and *Paenarthrobacter* LAR21 form AMPO without a strong decrease in MBOA, suggesting the existence of multiple ways to form AMPO from

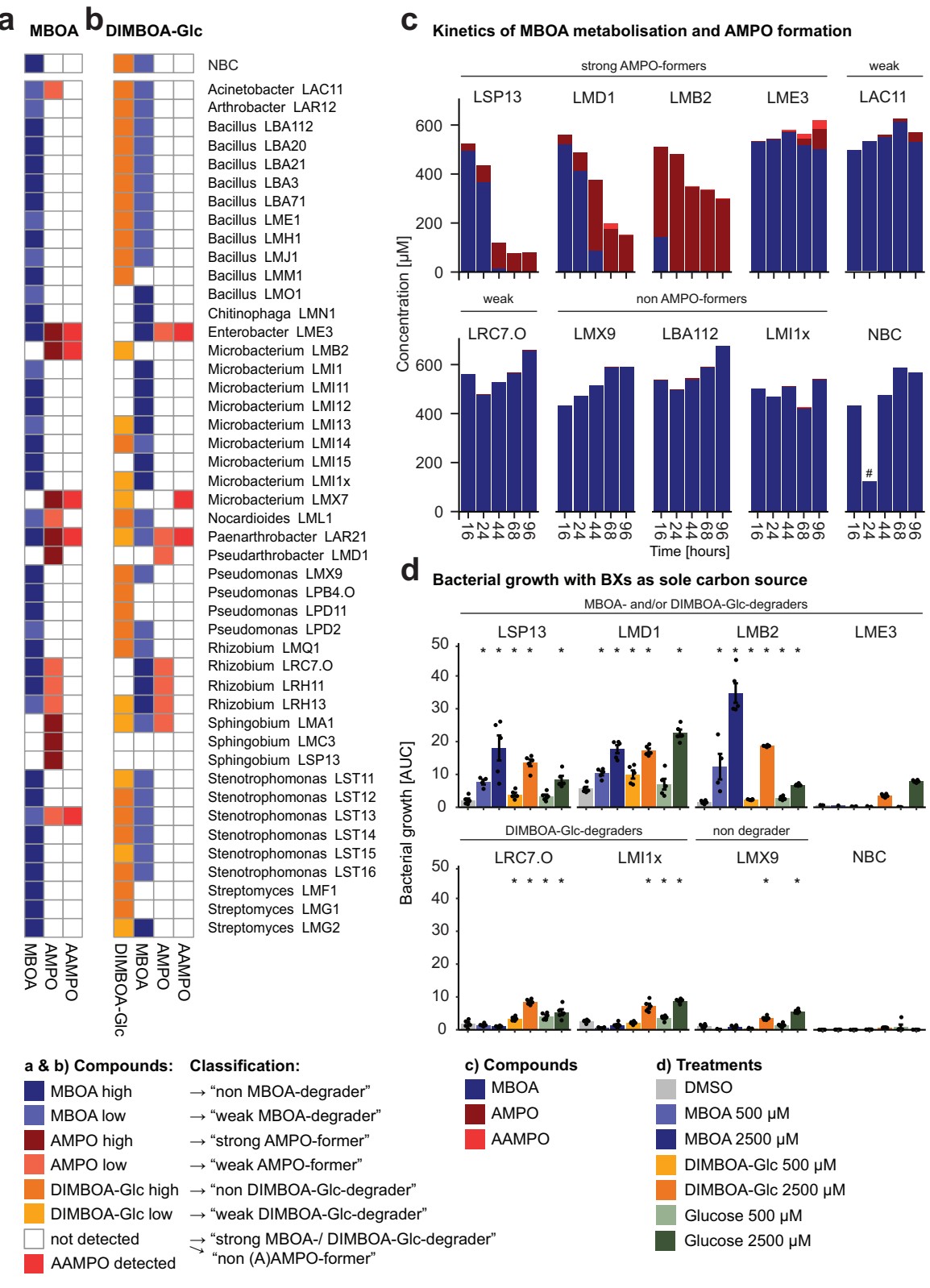

**a** MBOA  **b** DIMBOA-Glc

**c** Kinetics of MBOA metabolisation and AMPO formation

**d** Bacterial growth with BXs as sole carbon source

a & b) Compounds:
- MBOA high → "non MBOA-degrader"
- MBOA low → "weak MBOA-degrader"
- AMPO high → "strong AMPO-former"
- AMPO low → "weak AMPO-former"
- DIMBOA-Glc high → "non DIMBOA-Glc-degrader"
- DIMBOA-Glc low → "weak DIMBOA-Glc-degrader"
- not detected → "strong MBOA-/ DIMBOA-Glc-degrader" "non (A)AMPO-former"
- AAMPO detected

c) Compounds
- MBOA
- AMPO
- AAMPO

d) Treatments
- DMSO
- MBOA 500 µM
- MBOA 2500 µM
- DIMBOA-Glc 500 µM
- DIMBOA-Glc 2500 µM
- Glucose 500 µM
- Glucose 2500 µM

MBOA. The liquid assay confirmed the AMPO-formers, previously classified on plates (Fig. 1a), and allowed to conclude that strong MBOA-degraders were strong AMPO-formers.

To characterise the kinetics of AMPO formation and MBOA degradation, we next performed a time series experiment with nine strains varying in these traits. These strains included four strong (*Sphingobium* LSP13, *Pseudoarthrobacter* LMD1, *Microbacterium* LMB2,

and *Enterobacter* LME3) and two weak AMPO-formers (*Acinetobacter* LAC11 and *Rhizobium* LRC7.O) alongside three non-AMPO-formers (*Pseudomonas* LMX9, *Bacillus* LBA112 and *Microbacterium* LMI1x). Rapid AMPO formation was coupled with a strong decrease of MBOA (LSP13, LMD1 and LMB2) while low amounts of AMPO formed with time and without much decrease of MBOA (LME3 and LAC11; Fig. 2c). Neither MBOA degradation nor AMPO formation were detected in

**Fig. 2 | Metabolisation of benzoxazinoids and use as sole carbon source by maize root bacteria. a, b** Heatmaps displaying qualitative detection of MBOA, DIMBOA-Glc and their metabolisation products for 46 maize root bacteria. Detected compounds and classifications are indicated in the legend. The corresponding metabolite analyses and classification cut-offs are shown in Supplementary Fig. 5. **a–c** Strains were grown in liquid 50% TSB supplemented with 500 μM of the respective chemical. "No bacteria control" NBC only contains the medium supplemented with the respective chemicals. **a** Detection of MBOA and its metabolisation products AMPO and AAMPO and **b** DIMBOA-Glc and its metabolisation products MBOA, AMPO and AAMPO. **c** Time course of MBOA metabolisation to AMPO and AAMPO for selected single strains: strong AMPO-formers *Sphingobium*

LSP13, *Pseudoarthrobacter* LMD1, *Microbacterium* LMB2, *Enterobacter* LME3; weak AMPO-formers *Acinetobacter* LAC11 and *Rhizobium* LRC7.O; non-AMPO-formers *Pseudomonas* LMX9, *Bacillus* LBA112 and *Microbacterium* LMI1x. Metabolite measurements ($n = 1$) were made on pools of three independently grown cultures (#: sample with failed pooling). **d** Bacterial growth within 68 h (reported as the area under the growth curve, AUC) for selected strains in minimal medium supplemented with DMSO (negative control), glucose (positive control), MBOA and DIMBOA-Glc each in two concentrations (500 μM or 2500 μM). Means ± standard error and individual data points are shown ($n = 5$). Growth curves are available in Supplementary Fig. 6. Asterisks indicate significant differences between treatment and DMSO control (pairwise *t*-test, $P < 0.05$).

LRC7.O and the negative controls. Together with Fig. 2a these experiments indicate different ways for MBOA degradation: (i) MBOA can degrade to other products than AMPO, which we could not detect with our analytical method, (ii) MBOA is slowly degraded and little AMPO is slowly formed and, (iii) MBOA is rapidly and completely degraded resulting in rapid AMPO formation and potentially other non-detected products.

## Strong MBOA-degraders can also degrade DIMBOA-Glc

Since maize bacteria are not first exposed to MBOA on roots, we analysed the metabolisation of DIMBOA-Glc, the main benzoxazinoid exuded by maize roots[11]. DIMBOA-Glc is not commercially available and was purified from maize (traces of other co-purified benzoxazinoids, Supplementary Fig. 5b). Analogous to the above, we classified the strains and considered only the clear phenotypes (see "Methods" section and Supplementary Fig. 5c for cut-offs). We identified half of the strains as non-degraders and 12 strains as strong DIMBOA-Glc-degraders (>90% degraded; Fig. 2b). The DIMBOA-Glc-degraders generally accumulated MBOA in their cultures, while only for few strains AMPO and/or AAMPO was detected (Supplementary Fig. 5c). Importantly, strong MBOA-degraders were also degraders of DIMBOA-Glc, while some DIMBOA-Glc-degraders did not degrade MBOA, revealing that these two traits are not necessarily coupled in maize root bacteria.

## AMPO-formers can grow on MBOA as the sole carbon source

Chemically, the formation of AMPO requires the degradation of MBOA or earlier in the pathway the degradation of DIMBOA-Glc (Fig. 2a, b). Therefore, we tested whether DIMBOA-Glc and/or MBOA degrading strains could use these compounds as the sole carbon source and benefit in growth. We grew the same strains as above (Fig. 2c) in minimal media supplemented with either glucose, DIMBOA-Glc or MBOA at different concentrations. Most strains grew well in the positive control with glucose (Supplementary Fig. 6), except LAC11 and LBA112 that were then removed. The strong MBOA- and DIMBOA-Glc-degraders LSP13, LMD1, and LMB2 clearly had improved growth in the presence of both MBOA or DIMBOA-Glc (Fig. 2d). The other strains didn't grow on the tested MBOA concentrations but partially benefitted at higher DIMBOA-Glc concentration possibly making use of the glucose moiety. Together, these results reveal that the capacities to degrade DIMBOA-Glc and MBOA are associated with growth benefits under carbon-limiting conditions.

## AMPO formation varies within microbacteria

To identify the genetic basis of AMPO formation, we required a bacterial genus that includes both AMPO-forming and non-AMPO-forming strains. To expand the number of genomes and phenotypes beyond the MRB collection[21] (Fig. 1a), we also screened a selection of Arabidopsis bacteria of the AtSphere collection[38] for AMPO formation on MBOA-containing plates (see "Methods" section and Supplementary results). None of the tested microbacteria from Arabidopsis could form AMPO (Supplementary Fig. 7) while many microbacteria from maize did (Fig. 1a). Thus, for further analysis we selected the genus

*Microbacterium* and included all isolates from maize ($n = 18$) and Arabidopsis ($n = 17$), and additional four isolates from other plants that we had available in the laboratory (Supplementary Data 1; see "Methods" section). First, we confirmed this set of 39 microbacteria for AMPO formation using the plate assay and MBOA degradation and AMPO formation in liquid cultures (qualitative data in Fig. 3, quantitative metabolite data in Supplementary Fig. 8). MBOA was degraded and AMPO accumulated in cultures of most microbacteria classified as AMPO-formers in the plate assay. Even though AMPO formation was observed on the plate assay, no AMPO was detected in liquid cultures for three genomically similar strains (LTA6, LWH12, LWO13). The testing of MBOA metabolisation in liquid culture further uncovered four partially related strains (LWH10, LBN7, LWH11 and LWO12) that also accumulated HMPAA, an alternative degradation product of MBOA (Supplementary Fig. 1). Third, testing of DIMBOA-Glc metabolisation revealed that also non-AMPO-formers degraded DIMBOA-Glc and that most AMPO-forming strains were strong DIMBOA-Glc-degraders. An exception of the latter observation was a group of four genomically similar strains that formed AMPO following weak DIMBOA-Glc degradation (LM3X, LMB2, LMX7 and LWO14). Fourth, the 39 microbacteria were tested for growth in minimal media containing MBOA as the sole carbon source. AMPO-formers, except LWH11, grew on MBOA as the sole carbon source, corroborating that AMPO-forming strains have a growth benefit from this trait. Overall, the chemical validation provides a robust basis for comparative genomics of 16 AMPO-forming and 23 AMPO-negative *Microbacterium* strains.

## Identification of a gene cluster for AMPO formation in microbacteria

To identify candidate genes for AMPO formation, we used the AMPO-phenotype of the plate assay and combined three comparative genomic approaches. First, we compared the 39 genomes using OrthoFinder[39] and identified five orthogroups unique and specific to AMPO-forming strains (Fig. 3). While the orthogroups OG0002971, OG0002972, and OG0002973 contained single copy genes, OG0002149 and OG0001787 were present in varying copy numbers (Supplementary Data 2). Second, we screened the 39 genomes for short sequence strings associating with AMPO formation using a custom kmer approach (see "Methods" section). We identified 17 genes in *Microbacterium* LMB2 that contained high-scoring kmers in AMPO-formers (score ≥ 7 across all genomes) and showed significant associations (Fisher's exact test, $p < 0.05$) with the phenotype (Supplementary Data 3). Third, we analysed the transcriptional response of the AMPO-forming *Microbacterium* LMB2 to MBOA (see "Methods" section, Supplementary Fig. 9). The transcriptome analysis revealed 2.9% of genes being differentially regulated (108 genes) with 7 down- and 101 upregulated (Supplementary Data 4).

In summary, the orthogroup method identified 6 candidate genes (Fig. 4, Supplementary Table 1, Supplementary Data 2), the kmer approach resulted in 17 candidate genes (Supplementary Data 3) and the transcriptome analysis found 108 candidate genes in *Microbacterium* LMB2 (Supplementary Fig. 9, Supplementary Data 4). Three genes were identified with all three approaches, eight genes by the

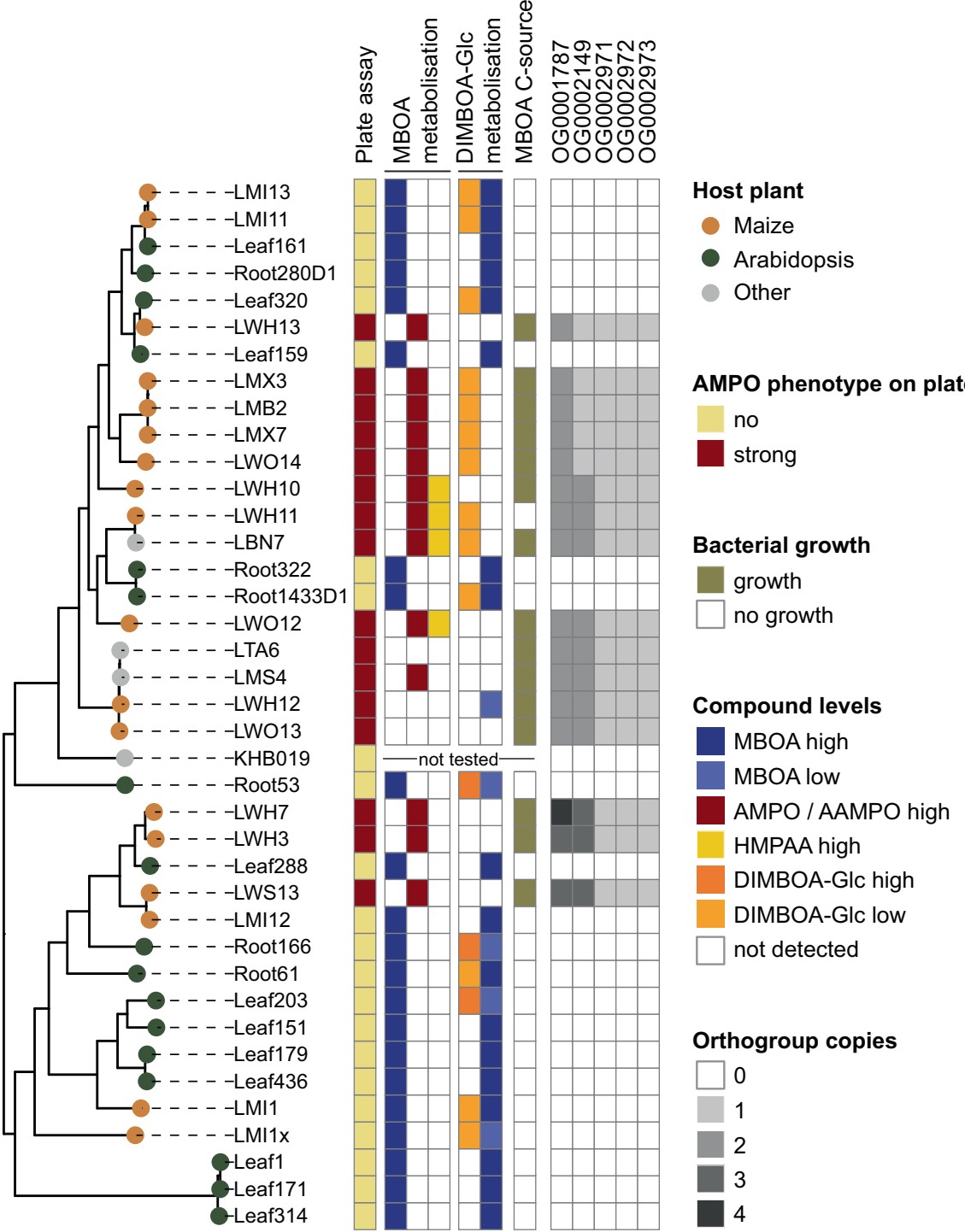

**Fig. 3 | Phenotypic diversity of AMPO-formation in microbacteria.** Phylogenetic tree constructed from whole genome alignment of 39 microbacteria. Tips are coloured by the host plant from which the strains were isolated. The column "plate assay" shows the AMPO classification (strong AMPO-former or non-AMPO-former) of the strains based on red colour formation on 100% TSA plates supplemented with MBOA (200 µg/mL). The adjacent columns display the classifications of metabolite analyses (MBOA, AMPO, HMPAA and DIMBOA-Glc) of liquid 50% TSB cultures supplemented with 500 µM MBOA ("MBOA metabolization") or DIMBOA-Glc ("DIMBOA-Glc metabolization") after 68 h. The column "MBOA C-source" refers to the assay where the strains were grown in minimal medium supplemented with 500 µM MBOA as a sole carbon source (based on mean results of 12 independent replicates grown in two independent runs). Columns "OG000xxxx" report copy numbers of gene orthogroups (Supplementary Data 2) that are unique and specific to AMPO-formers.

kmer and the RNAseq approach, and six genes by the orthogroup and the RNAseq approach (Fig. 4a). We selected the LMB2 genome as a reference because this strain was used also for the transcriptome analysis. Mapping the resulting candidates to the genome of LMB2 revealed 15 genes that were located adjacently, pointing to a gene cluster for AMPO formation (Fig. 4b). This gene cluster contained all six genes of the orthogroup analysis and all eight genes detected by the kmer and the RNAseq approach. Transcripts of the entire gene cluster were significantly upregulated in the presence of MBOA, corroborating an active role in MBOA degradation and AMPO formation

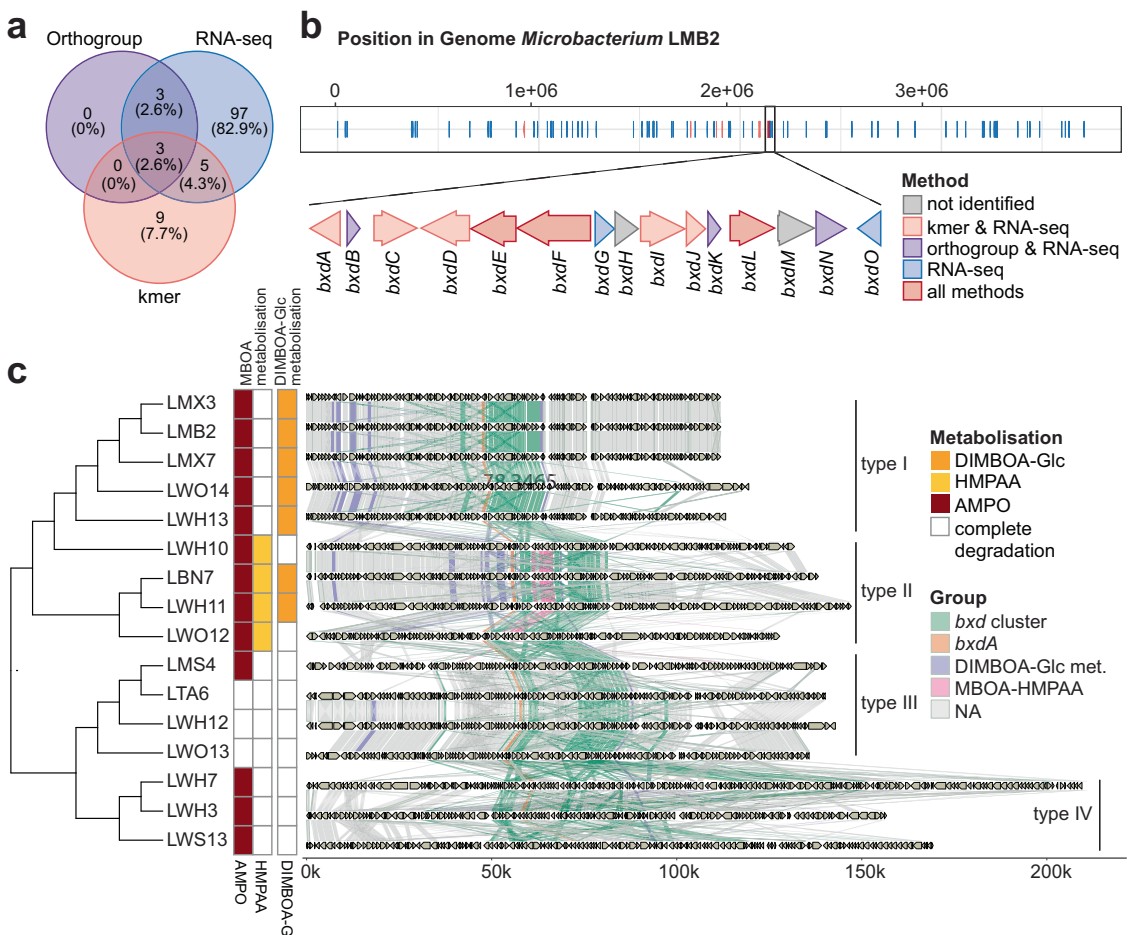

**Fig. 4 | Bxd gene cluster in microbacteria. a** Overlap of candidate genes for AMPO formation identified by three approaches: orthogroups, kmers and RNAseq. **b** Position of identified candidate genes in the genome of *Microbacterium* LMB2. A zoom-in of the benzoxazinoid degradation (*bxd*) gene cluster, annotated with its gene architecture including all genes named *bxdA* to *bxdO* is shown. **c** Synteny plot of the genomic region across microbacteria samples where the *bxd* gene cluster is found. The gene cluster can be categorized into four types (type I, type II, type III and type IV) based on their gene order, gene content and chemical phenotype.

(Supplementary Fig. 9). We termed this cluster **b**enzo**x**azinoid **d**egradation and named the 15 genes in sequence *bxdA* to *bxdO*. The *bxd* gene cluster encodes 13 enzymes and two transcriptional regulators (Supplementary Table 1).

To further compare the *bxd* gene clusters, we performed in-depth analysis of closed long-read genomes of all AMPO-forming microbacteria. High-resolution alignments revealed four types of cluster architectures (Fig. 4c). These clusters partially agreed with the different metabolisation phenotypes of the strains (Fig. 3). Interestingly, cluster type II contained five additional genes in the *bxd* gene cluster was found in four strains (LWH10, LWH11, LBN7, LWO12), that uniquely formed HMPAA besides accumulating AMPO. This fine-grained genome analysis suggests multiple variants of the *bxd* gene cluster, possibly representing multiple metabolic pathways of benzoxazinoid degradation in microbacteria.

### BxdA converts MBOA to AMPO in vitro

To identify the gene(s) responsible for MBOA breakdown and AMPO formation, we heterologously expressed four candidate genes in *E. coli*. We selected *bxdA*, *bxdD*, *bxd*G, and *bxdN* based on the functional annotation of their proteins as N-acyl homoserine lactonase family protein (BxdA), aldehyde dehydrogenase family protein (BxdD), VOC family protein (BxdG), and NAD(P)-dependent oxidoreductase (BxdN). While neither purified BxdD, BxdG and BxdN nor the empty vector control showed MBOA degrading activity (Supplementary Fig. 10), purified BxdA degraded MBOA and led to the accumulation of AMPO

(Fig. 5a). Hence, the *Microbacterium* gene *bxdA* encodes a ~34 kDa protein, annotated as an N-acyl homoserine lactonase family protein, that has in vitro activity to degrade MBOA and form AMPO.

### *Sphingobium* mutant Δ3bxdA lacks AMPO formation

Complementary to in vitro confirmation, we aimed at confirming the function of BxdA in vivo. Because microbacteria are genetically not amenable, we searched BxdA homologues in the strong AMPO-former *Sphingobium* LSP13. A protein blast search revealed three homologues in this strain with similarities to LMB2 BxdA ranging from 62–66% (MRBLSP13_002227, MRBLSP13_002921 and MRBLSP13_003006). We created a markerless triple deletion mutant (see "Methods" section), hereafter called Δ3bxdA, and tested it for AMPO formation. No colour change to red was observed for Δ3bxdA on MBOA-containing plates suggesting that the mutant fails to form AMPO (Fig. 5b). Also, in complex liquid media, Δ3bxdA did not form detectable levels of AMPO (Fig. 5c). However, the triple mutant could still degrade MBOA in complex medium albeit slower compared to the LSP13 wild-type strain. Thus, we wondered whether BxdA was required for growth on MBOA as the sole carbon source. Unlike the wild-type, the mutant Δ3bxdA failed to degrade MBOA (Fig. 5d) and was severely impaired in growth in minimal medium with MBOA as a sole carbon source (Fig. 5e, Supplementary Fig. 11a). These experiments reveal that BxdA is required for degradation of MBOA and AMPO formation in vivo. Furthermore, the slow degradation of MBOA by Δ3bxdA in complex medium indicates that *Sphingobium* LSP13 possesses another, BxdA-independent

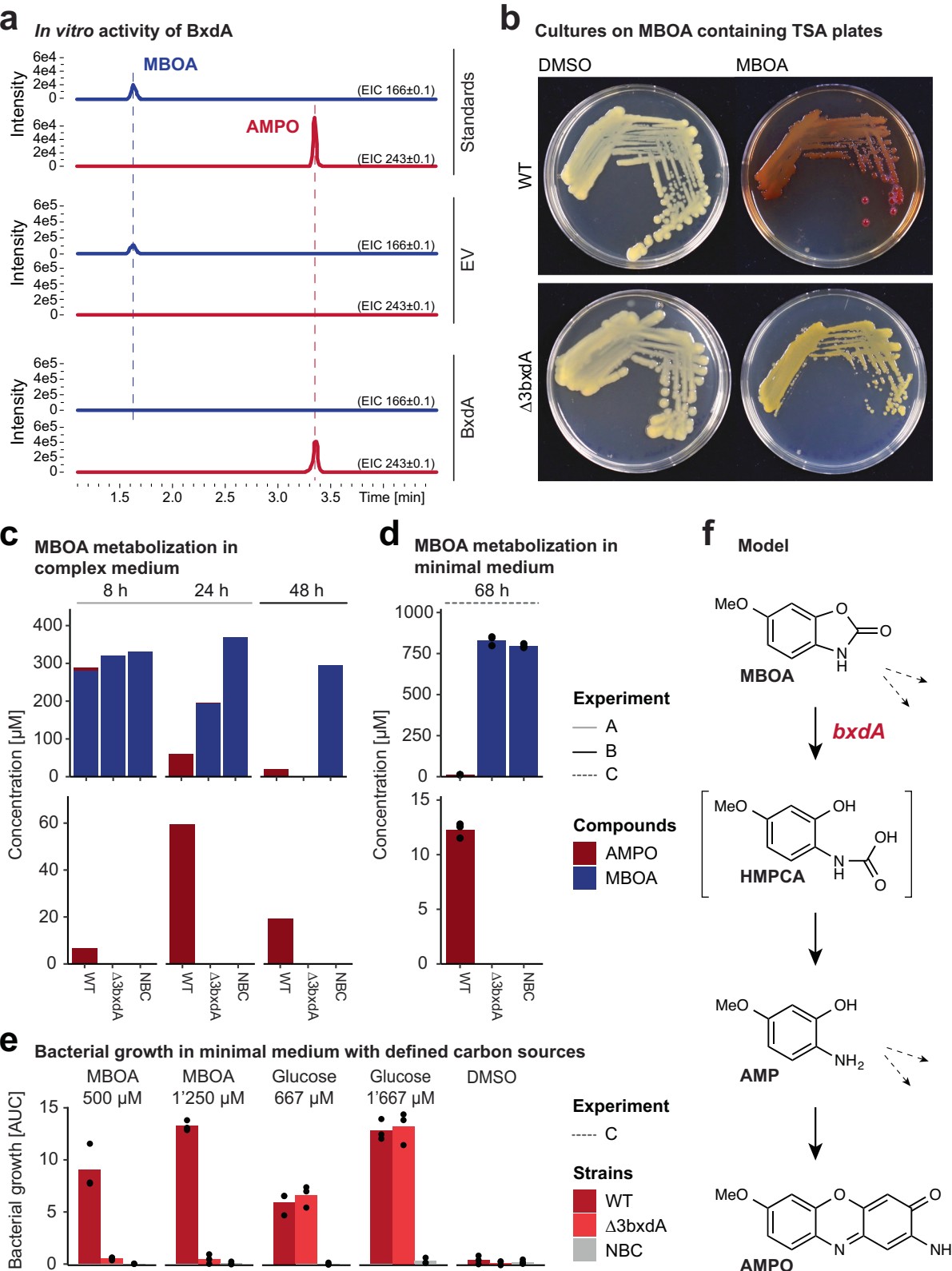

degradation pathway of MBOA that does not lead to the accumulation of AMPO. We propose that BxdA functions as a lactonase, opening the lactone moiety of MBOA to form 2-amino-5-methoxyphenol (AMP) via the corresponding carbamate (HMPCA) as potential intermediate and two molecules of AMP then react to AMPO (Fig. 5f). We expanded these findings of MBOA to apply largely to the non-methoxylated BX analogues BOA and APO, too (Supplementary Fig. 11, see Supplementary Results). Taken together the results demonstrate that BxdA degrades

MBOA and AMPO is formed not only in the microbacteria belonging to the Actinobacteriota but also in another phylum, the Pseudomonadata, namely the strain *Sphingobium* LSP13.

**Homology searches: BxdA is present in AMPO-forming maize root bacteria**

After identification of the *bxdA* gene encoding the N-acyl homoserine lactonase enzyme that is required for AMPO formation in

**Fig. 5 | BxdA converts MBOA to AMPO. a** In vitro activity of purified recombinant BxdA (from LMB2) with the substrate MBOA. Extracted ion chromatograms (EIC, HPLC-MS analysis in positive mode) for MBOA and AMPO are shown. An empty vector (EV) control and MBOA and AMPO standards were included. **b**–**e** Characterization of *Sphingobium* LSP13 Δ3bxdA mutant lacking the three *bxdA* homologs. **b** Photograph of LSP13 wild-type (WT) and the Δ3bxdA mutant cultivated for 10 days on 100% TSA supplemented with 200 μg/mL MBOA or DMSO. **c** MBOA metabolisation to AMPO over time in 50% TSB supplemented with 500 μM MBOA. The 8 and 24 h timepoints (measurements (*n* = 1) were pools of three independently grown cultures) and the 48 h timepoints (measurements (*n* = 3) were biological replicates) come from different experiments. **d** MBOA metabolisation to AMPO in minimal medium supplemented with 500 μM MBOA. Means and individual data points are shown (*n* = 3). **e** Bacterial growth within 94 h (reported as the area under the growth curve, AUC) in minimal medium supplemented with DMSO (negative control), glucose (positive control) and MBOA (500 μM or 1250 μM). Means and individual data points of three biological replicates are shown (*n* = 3). Growth curves are available in Supplementary Fig. 11. **f** Proposed reaction sequence from MBOA to AMPO catalysed by BxdA. The dashed arrows refer to possible alternative MBOA degradation pathways. The potential intermediate HMPCA ((2-hydroxy-4-methoxyphenyl)carbamic acid) is proposed but was not confirmed experimentally.

*Microbacterium* LMB2 and *Sphingobium* LSP13, we investigated how widespread and similar this gene is in other bacterial genomes. We assessed the *bxdA* sequence of the strain LMB2 (MRBLMB2_002078) in the genomes of the other *Microbacterium* spp. used in this study, in the available genomes of the other strains of the MRB collection[21], and in publicly available bacterial genomes (see "Methods" section). To identify homologues, we blasted the amino acid sequence and quantified their similarities.

Among the microbacteria tested in this study, all AMPO-forming strains possessed homologous BxdA proteins with high sequence similarities ranging from 76.3–100% (Supplementary Fig. 12a). The corresponding protein was missing in AMPO-negative strains, or the best hits showed low similarity (<25%). Among other strains of the MRB collection, homologues of the lactonase BxdA were missing or had low sequence similarity (<25%) in most non-AMPO-forming strains (Supplementary Fig. 12b). Consistent with the AMPO-forming phenotype, we found BxdA homologues with higher similarities in *Pseudoarthrobacter* LMD1 and *Sphingobium* LSP13, LMA1 and LMC3. The gene variants of *Pseudoarthrobacter* and *Sphingobium* (the 3 gene copies) showed amino acid sequence similarities of 78.9% and 58.9–64.9%, respectively. In *Pseudoarthrobacter* the *bxd* gene cluster is present in a similar organization as cluster type I in LMB2. This genotype is consistent with the chemical phenotype of *Pseudoarthrobacter* LMD1 with rapid MBOA degradation and AMPO formation (Fig. 2c). Hence, BxdA homologues of confirmed AMPO-forming strains have ~60% or higher amino acid sequence similarity.

Third, we searched BxdA homologues among the publicly available genomes of IMG[40]. Out of 123,000 genomes from isolates, we found *bxdA*-like genes in 462 genomes but only 28 had similarities >60% (amino acid sequence; Supplementary Fig. 12c, Supplementary Data 5). Most of these homologues were in genomes of bacteria isolated from plant and soil environments. *BxdA* genes were identified in bacteria of the Micrococcaceae family (an *Arthrobacter*), *Microbacterium* isolates and a *Sphingobium* isolate. We also identified *bxdA*-like genes in *Burkholderia* or different *Pseudomonas* isolates. Most of these taxa reflect the lineages that we confirmed to form AMPO with our isolate collection approach (Fig. 1a).

Finally, we compared BxdA from *Microbacterium* LMB2 with proteins previously reported to act in the metabolisation of benzoxazinoids. For instance, the metal-dependent hydrolase CbaA was identified in the bacterium *Pigmentiphaga*, an enzyme catalysing the degradation of a derivate of a benzoxazinoid to the corresponding aminophenoxazinone[33]. The metallo-ß-lactamases (mbl) of the fungus *Fusarium pseudograminearum* were found to degrade a benzoxazinoid[41]. BxdA only shared low 42.6% and 30.1% sequence similarity to CbaA and Mbl, respectively. Substrate specificities of BxdA, CbaA and Mbl, need to be investigated.

In summary, BxdA homologues are present in AMPO-forming strains. The high sequence similarity and the restricted presence in a few families indicate that this protein is rare and points to the specific association of BxdA-carrying bacteria with maize and the importance of BxdA for AMPO formation.

## Discussion

Plants recruit distinct root microbial communities from the soil by exuding bioactive specialised metabolites[5]. Thus, they shape species-specific microbiomes[5], but the mechanisms are not well understood. Here, we show that many maize root bacteria can metabolise host-exuded benzoxazinoids, the main specialised metabolites of maize. This trait is specific to native root bacteria from maize and is present among various genera and abundant members of the root microbiome. Metabolisation of benzoxazinoids was rare in root microbiomes of plants that do not produce benzoxazinoids, such as Arabidopsis, Brassica and Medicago (Fig. 1c). Maize bacteria benefitted from metabolising MBOA by using it as carbon source in nutrient limiting conditions. Of the different known chemical routes to degrade MBOA (Supplementary Fig. 1), we have identified BxdA, which encodes a putative lactonase enzyme required to convert the benzoxazinoid MBOA to the aminophenoxazinone AMPO. Through BxdA, maize root bacteria can metabolise the host-specific benzoxazinoids of maize. Below, we discuss the metabolisation of host-specific compounds, the biochemistry of BxdA, and the biological context of these findings.

Root microbes metabolise specialised plant metabolites. For example, Arabidopsis root colonizing *Pseudomonas* metabolises triterpenes[30] and tomato root colonizing *Sphingobium* degrades tomatine[8]. Several soil microbes convert benzoxazinoids to various benzoxazinoid derivates[17,42]. Here, we characterised maize root bacteria that metabolise the dominant benzoxazinoid in the maize rhizosphere, MBOA to AMPO. We found AMPO-forming bacteria as abundant colonizers of maize roots (~7.7% of cultured root bacteria, Fig. 1b). AMPO formation was found in several genera of maize root bacteria belonging to the phyla Actinobacteriota (*Microbacterium* and *Pseudoarthrobacter*) and Pseudomonadata (*Sphingobium, Enterobacter, Rhizobium, Acinetobacter*; Fig. 1d). Compared to bacteria isolated from Arabidopsis[38], more maize bacteria[21] formed AMPO (Fig. 1d, Supplementary Fig. 7a), corroborating the results from the plating of root extracts where no AMPO-forming bacteria were detected in Arabidopsis root extracts. Metabolite profiling of *Microbacterium* spp. from maize and Arabidopsis revealed that only maize-derived isolates metabolised benzoxazinoids (Fig. 3), and genomic comparisons uncovered the *bxd* gene cluster, which was only present in AMPO-forming isolates (Fig. 4). The key gene *bxdA* was also present in other MBOA-metabolising maize bacteria but not in Arabidopsis-derived bacteria (Supplementary Fig. 12), which suggests metabolic adaptation of maize root bacteria to host-specialised metabolites at the genomic level.

In line with this, roots of benzoxazinoid-deficient *bx1* mutants harboured 50% less AMPO-forming bacteria, highlighting a direct link between benzoxazinoid exudation from maize and bacterial MBOA metabolisation (Fig. 1b). Benzoxazinoid-metabolising bacteria were also enriched in the maize rhizosphere compared to root extracts of different plant species (Fig. 1c). The finding that AMPO-forming bacteria were less abundant in the wheat root microbiomes is consistent with the MBOA levels in the rhizosphere of the wheat variety tested. More than 10x less MBOA (~5 ng/mL) was found in the rhizosphere of

this wheat variety compared to maize (~60 ng/mL)[43]. It is probable that lower levels of MBOA in the rhizosphere of this wheat line resulted in a much weaker selection of AMPO-formers.

To investigate benzoxazinoid metabolisation, we focused on maize root bacteria metabolizing MBOA, the most abundant[11] and most selective[21] benzoxazinoid in the maize rhizosphere. The phenotypic and genomic screening of maize- and Arabidopsis-derived microbacteria (Fig. 3) permitted the identification of BxdA. Gene homologues were only found in AMPO-forming *Pseudoarthrobacter* and *Sphingobium* strains from maize but not in Arabidopsis bacteria (Supplementary Fig. 12). We detected weak similarity (<43% amino acid level) with known enzymes such as CbaA[33] or MBL[41], both involved in metabolisation of benzoxazinoids through aminophenol intermediates. Thus, the lactonase BxdA represents a key enzyme for benzoxazinoid metabolisation pointing to a highly specific adaption restricted to root microbiome members of benzoxazinoid-producing plants.

We confirmed that BxdA is sufficient to catalyse the metabolisation of MBOA to AMPO in vitro and that BxdA is required for AMPO formation by *Sphingobium* LSP13 in vivo (Fig. 5). The biochemistry of BxdA is consistent with its annotation as a lactonase that hydrolyses the ester bond of a lactone ring[44]. With MBOA as a substrate, this reaction yields AMP that spontaneously dimerizes to AMPO in the presence of oxygen[26] (Fig. 5f). Lactonases occur in various bacteria[45] and typically degrade N-acyl homoserine lactones, which are signalling metabolites of bacterial quorum sensing[46]. This supposedly similar biochemical function opens a range of follow-up questions, including on the evolutionary origin of BxdA or its impact on quorum sensing that warrant further investigation. Beyond *bxdA*, the role and contributions of the other genes in the *bxd* gene cluster to BX metabolisation remain to be uncovered.

Metabolisation of specialised metabolites has multiple biological consequences. Generally, bacteria aim at detoxification, suppression of other microbes, or utilization as carbon source[47,48]. Our analyses suggested that the bacteria primarily degrade MBOA (Fig. 2c), which is consistent with strong AMPO-formers using MBOA as a carbon source (Fig. 2d and 3). It is conceivable that AMPO is rather formed as a side product of incomplete or inefficient bacterial catabolism of the intermediate AMP, which is in line with the rather low levels of AMPO measured relative to the degraded MBOA. Production of the aminophenoxazinone side product APO has been found for the fungal pathogen *Fusarium verticilloides* when the transformation of the aminophenol intermediate in BOA detoxification was hindered or reduced[32]. Nevertheless, AMPO formation will affect the microbial and plant ecology. It confers advantages to AMPO-tolerant bacteria to expand their niche by preferentially suppressing Gram-positive bacteria, which are generally less tolerant to aminophenoxazinones compared to Gram-negative bacteria[21]. Alternatively, AMPO may promote rhizosphere health through its suppressive activity against phytopathogenic fungi[32,49]. Plants, on the other hand, may benefit from recruiting A(M)PO-forming bacteria as they convert (M)BOA to strongly allelopathic compounds that suppress weeds[50], thereby improving host fitness. Overall, AMPO-forming bacteria contribute to microbiome traits that benefit their host plant.

In our previous work, we found that maize root bacteria tolerate the specialised metabolites of their host[21] and here we show that they are adapted to metabolize the maize compounds. Together this reveals that the bacterial traits of tolerating and metabolising the specialised metabolites of their host are important determinants of maize microbiome membership. The next step will be to study the *in planta* functions of these bacterial traits for learning their effects on the maize host. Given the high degree of host species-specific microbiomes[5] and the widespread nature of plant species-specific specialised metabolites, we propose that metabolisation of host-specific specialised metabolites contributes to structuring root microbiomes across the plant kingdom. Regarding possible agricultural applications, our data implies that effective biocontrol or biofertilizer strains should be tolerant and/or metabolically adapted to the specialised metabolites produced by the target crop. Hence, understanding how specific specialised plant metabolites shape and stabilize their microbiomes will be important to harness microbiome functions to improve plant health in sustainable agricultural systems[51].

## Methods

### Plating experiment

To assess the number of AMPO-forming colonies on roots, we grew wild-type B73 maize plants and BX-deficient *bx1*(B73) maize, wheat (CH Claro), *Medicago sativa* (Sativa, Rheinau, Switzerland), *Brassica napus* (Botanik Saemereien AG, Pfaeffikon, Switzerland) and *Arabidopsis thaliana* (Col-0) in field soil. The soil was collected in the Winter of 2019 from the field in Changins[11]. We grew the plants for 7 weeks in a walk-in growth chamber with the following settings: 16:8 light/dark, 26/23 °C, 50% relative humidity, ~550 µmol m$^{-2}$s$^{-1}$ light. We fertilized the plants in the following regime: Weeks 1–4: 100 mL; 0.2% Plantactive Typ K (Hauert HBG Duenger AG, Grossaffoltern, Switzerland), 0.0001% Sequestrene Rapid (Maag, Westland Schweiz GmbH, Dielsdorf, Switzerland); weeks 5 onwards: 200 mL; 0.2% Plantactive Typ K, 0.02% Sequestrene Rapid. To account for the different needs of Arabidopsis growth, all seeds were stratified for three days in the dark at 4 °C and then grown in growth cabinets (Percival, CLF Plant climatics) at 60% relative humidity, 10 h light at 21 °C and 14 h dark at 18 °C. Arabidopsis were fertilized two times during the experiment by watering with 2/3 water and 1/3 of 50% Hoagland solution[52]. To harvest the roots, we shook off loose soil and prepared 10 cm long root fragments (corresponding to the depth of −1 to −11 cm in soil) which we then chopped into small pieces with a sterile scalpel. We transferred them into a 50 mL Falcon tube containing 10 mL sterile magnesium chloride buffer supplemented with Tween20 (MgCl$_2$Tween, 10 mM MgCl$_2$ + 0.05% Tween, both Sigma–Aldrich, St. Louis, USA). We homogenized the roots with a laboratory blender (Polytron, Kinematica, Luzern, Switzerland; 1 min at 20,000 rpm) followed by additional vortexing for 15 s. For the rhizosphere fraction, we resuspended the pellet from the washing step in 5 mL MgCl$_2$Tween. For the soil fraction, we mixed 5 g of soil from the pot with 5 mL MgCl$_2$Tween and vortexed it for 15 s.

To quantify bacterial community size, we plated root, rhizosphere, and soil extracts. We serially diluted the extracts and plated 20 µL on 10% TSA (3 g/L tryptic soy broth and 15 g/L agar, both Sigma–Aldrich, St. Louis, USA) plates (12 × 12 cm, Greiner bio-one, Kremsmünster, Austria) containing filter-sterilized cycloheximide (10 mg/L, Sigma–Aldrich, St. Louis, USA) and filter-sterilized DMSO (2 mL/L, Sigma–Aldrich, St. Louis, USA). To spread the drops for counting we tilted the plates and incubated them for 6 days at room temperature. We counted colony-forming units (CFU), multiplied them by the dilution factor and normalized them with the sample's fresh weight. Before statistical analysis, we transformed CFU counts by log10.

To count the number of AMPO-forming colonies in the extracts, we spread one dilution on a square agar plate containing MBOA. Depending on the plant species and the compartment, we selected a dilution between 1:10$^{-1}$ and 1:10$^{-4}$ to reach a colony density which is countable. We spread 50 µL of the sample with a delta cell spreader on square agar plates containing 10% TSA supplemented with filter-sterilized cycloheximide and filter-sterilized MBOA (200 mg/L, Sigma–Aldrich, St. Louis, USA). The MBOA concentration in the plates corresponds to ~1200 µM. For 10 days we incubated the plates at room temperature (21–25 °C). We photographed the plates and counted the red colonies on the pictures. To get the proportion of AMPO-forming colonies per sample, we divided the count of AMPO-forming colonies by the total CFU.

## Bacterial strains and growth conditions

Maize root bacteria (i.e., MRB collection)[21] and Arabidopsis bacteria (i.e., AtSPHERE collection)[38] were routinely grown on solid 100% TSA plates (30 g/L tryptic soy broth and 15 g/L of agar, both Sigma−Aldrich) at 25 °C–28 °C or in liquid 50% TSB medium (15 g/L tryptic soy broth). For the cryo stocks, bacteria were grown for 48 h in liquid 100% TSB (30 g/L tryptic soy broth) and mixed with sterile-filtered glycerol (Sigma−Aldrich) at a final concentration of 20%. Most strains of the MRB collection were initially screened for AMPO formation on MBOA-containing plates (Fig. 1d) and a taxonomically broad subset of 50 strains was selected for metabolisation analysis in liquid culture (Fig. 2a, b). Criteria for selection were (i) to have representatives of the different families, (ii) differing 16S rRNA sequences within a lineage and (iii) good growth in the test conditions in liquid TSB medium.

We also screened Arabidopsis bacteria for AMPO formation on MBOA-containing plates. These strains originated from the AtSphere collection[38] and have publicly available genomes. Our rational was to expand the diversity and number of strains to have paired information of genomes and AMPO formation phenotypes for subsequent comparative genomics. We did not aim for a systematic comparison of the two collections but selected the Arabidopsis bacteria with high 16S rRNA gene sequence similarity to the strains of the MRB collection. This resulted in a selection of 57 strains that were isolated either from Arabidopsis roots or leaves or from the soil where Arabidopsis was grown to build the AtSphere collection.

## Plate-based screen for AMPO formation

To screen for AMPO formation of single isolates, we plated a loop of pure bacterial cultures on 100% TSA plates supplemented with ~1200 μM MBOA (200 mg/L) or DMSO (2 mL/L) as control. We incubated the plates for 10 days at ambient temperature, assessed the phenotype by eye and photographed the plates. Based on the colour of the colony and surrounding media, we qualitatively classified the strains for their phenotype of AMPO formation as strong, weak or non-AMPO-formers. Strong AMPO-formers provoke a clear colour change to (dark) red on MBOA plates, while light red colouration is characteristic for weak AMPO-formers. No colour change is detected on MBOA plates compared to the DMSO control plates for non-AMPO-formers. Supplementary Fig. 3 shows examples of this classification scale.

## Abundance estimation of AMPO-formers

Mappings of 16S Sanger sequences of the MRB strains to relative abundances of culture-independent 16S amplicon sequencing data were used from our previous study[21] and processed as follows. Briefly, the relative abundances of a strain were estimated by summing the relative abundances of all ASVs (amplicon sequence variants) to which the 16S Sanger sequence of the strain showed a similarity >99% in a given dataset. First, we used the mapping to the bacterial community data of the roots from which the MRB strains were isolated from[11] (i.e., wild-type B73 maize in the greenhouse experiment). Second, to estimate the cumulative relative abundance of AMPO-formers in field data[9] (i.e., maize root samples of B73 in Reckenholz and Changins fields and W22 in Aurora), relative abundances of all ASVs that were mapped at 99% similarity by at least one AMPO-former were summed up the same way.

## Liquid culture assays and experiments

We utilized our previously described 96-well liquid culture-based growth system[21,53] for several different assays in the presence and absence of different BX compounds. The different assays included bacterial growth measurements in complex and minimal media as well as metabolite or transcriptome analyses. The general setup was as follows: First, pre-cultures were prepared by transferring isolate colonies with inoculation needles (Greiner bio-one, Kremsmünster, Austria) to 1 mL of liquid 50% TSB (15 g/L tryptic soy broth,

Sigma−Aldrich) in 2 mL 96-well deep-well plates (Semadeni, Ostermundigen, Switzerland). These pre-culture growth plates were covered with a Breathe-Easy membrane (Diversified Biotech, Dedham, USA) and grown until stationary phase for 4 days at 28 °C and 180 rpm.

Then 4 μL of the pre-cultures were inoculated to 200 μL fresh liquid 50% TSB in 96-well microtiter plates (Corning, Corning, USA) containing the compounds and concentrations to be tested (see below). Of note, we also tested DIMBOA, which is a main compound in maize exudates (Supplementary Fig. 1), however, we did not analyse it further because of spontaneous conversion to MBOA in the absence of bacteria in the assay. The chemical treatments were prepared by mixing their stock solutions into liquid 50% TSB. DIMBOA-Glc was isolated from maize seedlings as described previously[21] and has a purity of 70% (Supplementary Fig. 5b). Synthetic MBOA and BOA were commercially available (Sigma−Aldrich). Stock solutions were prepared in the solvent DMSO (Sigma−Aldrich) depending on the solubility of the compounds: DIMBOA-Glc at 500 mM (187 mg/mL), MBOA at 606 mM (100 mg/mL) and BOA at 500 mM (68 mg/mL). The DMSO concentration was kept constant in each treatment including the controls. In each plate, wells with 50% TSB were included as no bacteria controls (NBC) and in each run one plate containing only media was included to monitor potential contaminations. In the following, we describe the specific details of the different experiments performed with liquid assays.

1. Metabolite analysis of MRB strains: We tested a taxonomically broad set of 46 strains of the MRB collection for their capacity to metabolise MBOA and DIMBOA-Glc (Fig. 2a, b). Goals were to identify strains that can degrade MBOA (500 μM) and DIMBOA-Glc (500 μM) as well as to confirm the ones that form AMPO. The 96-well plates were grown under continuous shaking on a laboratory shaker (28 °C, 180 rpm shaking). To avoid evaporation, we sealed the plates with a Breathe-Easy membrane. After 68 h of growth, we stopped the experiment and recorded the optical density (OD600) of the cultures in a plate reader (Tecan Infinite M200 multimode microplate reader equipped with monochromator optics; Tecan Group Ltd., Männedorf, Switzerland). We then fixed the bacterial cultures for metabolite extraction and analysis of benzoxazinoid compounds and degradation products (see below). The obtained data is semi-quantitative as it was not normalized by bacterial cell numbers, and therefore we qualitatively classified the strains similar to the above assay with MBOA-containing plates. We classified a strain's capacity to degrade MBOA or DIMBOA-Glc as strong (>90% degraded relative to the levels detected in the control sample), weak (90 > × > 30% of the control) or non-degrading (<30% of the control). Analogously, we classified the strains as strong (>10% of max. AMPO-former), weak (<10% of max. AMPO-former) or non-AMPO-formers (<0.1% of max. AMPO-former). We recorded AAMPO formation but didn't classify the strains because the measured levels were very low. Supplementary Fig. 5 shows the metabolite data together with these cut-offs of the classification scale.

2. Transcriptome of LMB2: We quantified the transcriptomic response of the *Microbacterium* LMB2 in response to 500 μM MBOA (Fig. 4a and Supplementary Fig. 9). We used the same setup as for Experiment 1 except that bacterial cultures were grown for 16 h. Again culture densities and metabolites were analysed and additionally, we collected at harvest samples for transcriptome analysis. Six individual wells were pooled and immediately stabilized by the addition of RNAprotect Bacteria Reagent (Qiagen, Hilden, Germany). See below for the details on RNA extraction, sequencing and transcriptome analysis.

3. Time series: We performed a time series experiment with four strong (*Sphingobium* LSP13, *Pseudoarthrobacter* LMD1, *Microbacterium* LMB2, and *Enterobacter* LME3), two weak AMPO-formers (*Acinetobacter* LAC11 and *Rhizobium* LRC7.O) and three

non-AMPO-formers (*Pseudomonas* LMX9, *Bacillus* LBA112 and *Microbacterium* LMI1x) to characterise the kinetics of MBOA degradation and AMPO formation (Fig. 2c). In this experiment 500 μM of MBOA was used and bacterial growth in 96-well plates was monitored more high-throughput using a stacker (BioStack 4, Agilent Technologies, Santa Clara, United States), which was connected to a plate reader (Synergy H1, Agilent Technologies, Santa Clara, United States). Using this system, OD$_{600}$ of every culture was recorded every ~100 min during the time course. Prior to each measurement, the plates were shaken for 120 s. For metabolite analyses over time, we removed replicate plates from the stack after 16, 24, 44, 68 and 96 h. Fixing of bacterial cultures and metabolite analysis was performed as detailed below. Analogously, we have quantified MBOA metabolisation after 68 h of selected Arabidopsis strains in 500 μM MBOA (Supplementary Fig. 7b).

4. Bacterial growth in minimal media: We use the same strains of the time course experiment and we tested whether they could use DIMBOA-Glc and/or MBOA as sole carbon source for a benefit in growth (Fig. 2d). Using the stacker system, we followed the same procedure as described above but using minimal media instead of TSB to test for bacterial growth. The minimal media was prepared as described previously[54] and complemented with defined amounts of stock solutions of either MBOA or DIMBOA-Glc as sole carbon source to reach a final concentration of 500 or 2500 μM. As positive controls for growths, we grew the bacteria in glucose (500 and 2500 μM) as the sole carbon source.

5. Metabolisation and growth of *Microbacterium* spp.: To have a robust basis for comparative genomics, we confirmed the set of 39 *Microbacterium* strains from maize and Arabidopsis in the liquid culture settings for MBOA degradation, AMPO formation and growth on MBOA as the sole carbon source (Fig. 3). We used again the stacker system and followed the protocols (3) and (4) described in this section.

6. Metabolisation of BOA: The same set of strains of the time series experiment (3) was used to test whether strong MBOA degrading *Microbacterium* and *Sphingobium* strains would also degrade BOA (Supplementary Fig. 11b). The experiment was prepared as the time series experiment but with 500 μM BOA (and MBOA) and bacterial cultures were fixed at 68 h for metabolite analysis (details below).

7. *Sphingobium* LSP13 and Δ3bxdA mutant: We have also characterised the *Sphingobium* strain LSP13 and the corresponding mutant Δ3bxdA (see below) with the different assays in the stacker-based liquid growth system (Fig. 5c–e). For MBOA metabolisation in complex medium (50% TSB), we grew triplicate pre-cultures in 96-deep-well plates overnight, washed them once with 10 mM MgCl$_2$, adjusted the OD$_{600}$ to 1 and inoculated 200 μl main cultures in 96-well plates as described above. We tested 500 μM MBOA for metabolite analysis (details below). Plates were transferred to the stacker system described above and absorbance at 600 nm measured roughly every 15 min with 2 min of shaking prior to measurements. At 8 h and 24 h, individual plates were removed and cultures fixed for metabolite extraction. For metabolisation and growth determination at 48 h, we ran a separate experiment with only one 96-well plate. We closed the plate with a lid and sealed the sides with Breathe-Easy membrane. The plates were incubated continuously in the plate reader (Synergy H1, Agilent Technologies, Santa Clara, United States) over 48 h and absorbance at 600 nm was measured every 15 min with 2 min of shaking prior to measurements. At 48 h cultures were fixed for metabolite analysis as described below. For growth and MBOA metabolisation in minimal medium, triplicate pre-cultures were grown in 13 ml culture tubes (SARSTEDT, Nuembrecht, Germany) containing 3 ml of 50% TSB with constant

shaking. Overnight cultures were subcultured for 4 h and washed with minimal medium without carbon source. OD$_{600}$ was adjusted and main cultures were inoculated in a 96-well plate and sealed as described above. We tested MBOA and BOA concentrations of 500 and 1250 μM as sole carbon sources for growth and only the 500 μM condition was used for metabolite analysis (Fig. 5e and Supplementary Fig. 11; details below). As a control, glucose was used at 667 μM and 1667 μM corresponding to C equivalents of 500 μM and 1250 μM MBOA, respectively. Plates were incubated under continuous shaking in a plate reader (Synergy H1, Agilent Technologies, Santa Clara, United States) for 94 h and absorbance at 600 nm was measured every 10 min with shaking in between. For metabolite analysis, a separate 96-well plate was prepared as described but incubated under continuous shaking in a laboratory shaker. After 68 h, the plate was removed from the shaker, OD$_{600}$ measured using a plate reader (Synergy LX, BioTek, Winooski, United States) and cultures were fixed as described below.

General handling of stacker data: For all assays in the stacker-based high-throughput system, we exported bacterial growth data from the software (Gen 5, Agilent Technologies, Santa Clara, United States) to excel. We used R statistical software (version 4.0, R core Team, 2016) to analyse growth data. For a measure of total bacterial growth, we calculated the area under the growth curve (AUC of the x-axis for time and y-axis for OD$_{600}$) using the function *auc()* from package MESS[55] and normalized growth in treatment relative to the control. Such normalized bacterial growth data of a given concentration was statistically assessed (compound vs control) using one-sample *t*-tests (*p*-values adjusted for multiple hypothesis testing). See analysis scripts for further details. The general statistical analysis is described below.

## Metabolite extraction from bacterial cultures

To fix bacterial cultures, we added 150 μL bacterial cultures to 350 μL of the extraction buffer (100% methanol + 0.14% formic acid) in non-sterile round bottom 96-well plates (Thermo Fisher Scientific, Waltham, USA). We stored the fixed samples with a final concentration of 70% methanol and 0.1% formic acid at −80 °C. To reduce the number of samples, we pooled three replicates of the same culture. For the transcriptome experiment (*n* = 7 for bacterial samples and *n* = 3 for control samples) and for the metabolisation of *Sphingobium* LSP13 wild-type and the mutant Δ3bxdA in minimal medium (*n* = 3), we did not pool samples. We diluted the pooled samples by adding 50 μL to 700 μL MeOH 70% + 0.1% FA. We filtered the cultures through regenerated cellulose membrane filters (CHROMAFIL RC, 0.2 μm, Macherey−Nagel, Düren, Germany) by centrifugation (3220 × *g* for 2 min) to remove bacterial debris. To avoid any residual particles, we centrifuged the extracts at 11,000 × *g* for 10 min at 4 °C. We aliquoted the supernatants in glass vials (VWR, Dietikon, Switzerland) and stored the samples at −20 °C until analysis.

## Profiling benzoxazinoid degradation products in bacterial cultures

Using an Acquity I-Class UHPLC system (Waters, Milford, US) coupled to a Xevo G2-XS QTOF mass spectrometer (Waters, Milford, US) equipped with a LockSpray dual electrospray ion source (Waters, Milford, US) we quantified benzoxazinoids in samples of filtered bacterial cultures. Gradient elution was performed on an Acquity BEH C18 column (2.1 × 100 mm i.d., 1.7 mm particle size; Waters, Milford, US) at 98−50% A over 6 min, 50−100% B over 2 min, holding at 100% B for 2 min, re-equilibrating at 98% A for 2 min, where A = water + 0.1% formic acid and B = acetonitrile + 0.1% formic acid. The flow rate was 0.4 mL/min. The temperature of the column was maintained at 40 °C, and the injection volume was 1 μL. The QTOF MS was operated in sensitivity mode with a

positive polarity. The data were acquired over an m/z range of 50–1200 with scans of 0.1 s at a collision energy of 6 V (low energy) and a collision energy ramp from 10 to 30 V (high energy). The capillary and cone voltages were set to 2 kV and 20 V, respectively. The source temperature was maintained at 140 °C, the desolvation temperature was 400 °C at 1000 L/h and the cone gas flow was 100 L/h. Accurate mass measurements (<2 ppm) were obtained by infusing a solution of leucine encephalin at 200 ng/mL at a flow rate of 10 µL/min through the Lockspray probe (Waters, Milford, US). For each expected benzoxazinoid compound, four standards with concentrations of 10, 50, 200, and 400 ng/mL were run together with the samples (DIMBOA-Glc, DIMBOA, HMBOA, MBOA-Glc, MBOA, BOA, AMPO, APO, AAMPO, HMPMA) or 40, 200 ng/mL, 1 and 10 µg/mL for HMPAA and AMP.

## NMR identification of AMPO

To confirm the presence of AMPO in the liquid cultures of *Sphingobium* LSP13 and *Microbacterium* LMB2, we analysed them by $^1$H NMR spectroscopy (Bruker Advance 300, 1H: 300.18 MHz, Bruker Corp., Billerica, MA, USA). Briefly, liquid cultures were centrifuged (20 min, 11,000 g) and the supernatants extracted twice with $Et_2O$, dried with $Na_2SO_4$, and filtered in a glass funnel with cotton wool. During cultivation, a red precipitate formed towards the neck of the Erlenmeyer flasks, i.e., at the edge of the shaking cultures (Supplementary Fig. 2). This red precipitate left was collected from the Erlenmeyer flasks with acetone. The two extracts were combined, concentrated under reduced pressure, and dried over $P_2O_5$. The $^1$H NMR spectrum of the red residue obtained was recorded in DMSO-$d_6$ and compared to an analytical AMPO standard[25,56], confirming its presence in our bacterial cultures.

## Phylogenetic tree construction

The phylogenetic tree of all MRB and AtSphere bacteria was computed as described previously[21]. The species tree estimation for microbacteria was obtained from OrthoFinder v. 2.3.8[39]. The 16S trees were reconstructed as follows: First, the 16S sequences were combined into a single FASTA file and then aligned using MAFFT v. 7.475[57] with default options. The aligned sequences were then used as input to RAxML v. 8.2.12[58]. The multi-threaded version ' raxmlHPC-PTHREADS' was used with the options ' -f a -p 12345 -x 12345 -T 23 -m GTRCAT' with 1000 bootstrap replicates. The phylogenetic tree was visualized and annotated in R using the package ggtree[59].

## Comparative genomics

To find genes that are involved in the transformation of MBOA to AMPO we built an extended collection of 39 *Microbacterium* strains with paired information of available genomes as well as AMPO formation phenotypes. We selected 18 *Microbacterium* strains from maize[21] and 17 from the *AtSphere* collection[38] isolated from Arabidopsis and one strain isolated from clover[60]. Of note, the 18 *Microbacterium* strains from maize present the subset of MRB strains (the collection has a total of 42 *Microbacterium* strains) with sequenced genomes (available from BioProject PRJNA1009252 associated with the MRB collection[21]). For genome sequencing, *Microbacterium* strains that differed in their full-length 16S rRNA gene sequence had been selected (only one strain was sequenced among strains with identical full-length 16S rRNA gene sequence)[21]. Additionally, we included three strains which we isolated from root extracts of *Brassica napus* (LBN7), *Triticum aestivum* (LTA6) and *Medicago sativa* (LMS4) due to their red colony phenotype on MBOA plates (genomes available from BioProject PRJNA1083430). For those strains, we sequenced the genome by PacBio as described for the MRB collection[21]. The 39 microbacteria were phenotypically divided into AMPO-forming (*n* = 16) and AMPO-negative microbacteria (*n* = 23) strains based on the MBOA plate assay. Two approaches were investigated independently. The first consisted of grouping the genes into orthogroups with OrthoFinder v. 2.3.8[39] and

estimating significant associations between the phenotype and orthogroups by applying Fisher's Exact Test using the gene trait matching tool in OpenGenomeBrowser[61]. In the second approach, a kmer-similarity search strategy was conducted. The scaffolds of the assemblies were first divided into unique kmers of size 21 base pairs and counted using the tool Kmer Counter v. 3.1.1[62]. The resulting kmer libraries per sample were then merged into a single matrix using custom Python scripts. In the next step, the kmers were scored based on their occurrence in AMPO-positive or negative strains. Specifically, the score of a kmer was increased by 1, if the kmer is present in a sample with AMPO-forming phenotype and was decreased by 1 if the kmer is present in a sample with AMPO-negative phenotype. This score can thus be seen as a correlation between genetic sequence and phenotype. The highest-scoring kmers were then used to filter genes containing those kmers using custom Python scripts. Since this approach relies on exact matches of kmers, the gene sequences containing high-scoring kmers were clustered with a 70% similarity cut-off using vsearch v. 2.17.1[63]. The obtained centroid sequences were then searched with BLAST v. 2.10.0[64] against a database of all genes from all microbacteria strains using 'blastn'. The BLAST output was filtered for matches with an *e*-value < 1e50 which resulted in a list of genes for each centroid sequence. These gene lists were then statistically assessed for their association with the phenotype using Fisher's Exact Test in R (v. 4.2.1). The *p*-values were corrected using the Benjamini-Hochberg method.

## Transcriptome analysis

Bacterial cells were lysed by enzymatic lysis and proteinase K treatment and total RNA was extracted using the RNeasy Mini Kit (Qiagen, Hilden, Germany) with subsequent DNase treatment using the Rapid-Out DNA removal kit (Thermo Fisher Scientific, Waltham, USA) following manufacturer's instructions. The quantity and quality of the purified total RNA were assessed using a Thermo Fisher Scientific Qubit 4.0 fluorometer with the Qubit RNA BR Assay Kit (Thermo Fisher Scientific, Waltham, USA) and an Advanced Analytical Fragment Analyzer System using a Fragment Analyzer RNA Kit (Agilent, Basel, Switzerland), respectively. One hundred ng of input RNA was first depleted of ribosomal RNA using an Illumina Ribo-Zero plus rRNA Depletion Kit (Illumina, San Diego, US) following Illumina's guidelines. Thereafter cDNA libraries were made using an Illumina TruSeq Stranded Total Library Prep Kit (Illumina, San Diego, US) in combination with TruSeq RNA UD Indexes (Illumina, San Diego, US) according to Illumina's reference guide documentation. Pooled cDNA libraries were sequenced paired end using an Illumina NovaSeq 6000 SP Reagent Kit v1.5 (100 cycles Illumina, San Diego, US) on an Illumina NovaSeq 6000 instrument. The run produced, on average, 14 million reads/sample. The quality of the sequencing run was assessed using Illumina Sequencing Analysis Viewer (Illumina version 2.4.7) and all base call files were demultiplexed and converted into FASTQ files using Illumina bcl2fastq conversion software v2.20. The quality control assessments, generation of libraries and sequencing were conducted by the Next Generation Sequencing Platform, University of Bern. The raw data of the RNAseq experiment is available from NCBI's Gene Expression Omnibus (GEO) repository with the accession number GSE263275.

The quality of the RNA-Seq data was assessed using fastQC v. 0.11.7[65] and RSeQC v. 4.0.0 2[66]. The reads were mapped to the reference genome using HiSat2 v. 2.2.13[67]. The reference genome of strain LMB2 was prepared before the mapping step as follows: The General Features Format (GFF) file obtained from the assembly was transformed to the Gene Transfer Format (GTF) using AGAT v0.8.0[68] and subsequently transformed to Browser Extensible Data (BED) format using BEDOPS v. 2.4.39[69]. The HiSat2 index from the reference FASTA file was created using the 'hisat2-build' command. FeatureCounts v. 2.0.14[70] was used to count the number of reads overlapping with each gene as specified in the genome annotation. The Bioconductor

package DESeq2 (v1.32.0 5)[71] was used to test for differential gene expression between the experimental groups.

## Synteny of *bxd* gene cluster across microbacteria

In order to visualize the *bxd* gene cluster across different AMPO-forming *Microbacterium* strains, a synteny plot was created using the R packages gggenes[72] and gggenomes[73] and ggtree[59]. The gene links between strains were determined based on pairwise BLAST identities. The colouring of interesting genes was based on shared membership in significant orthogroups for a given phenotype as determined using Orthofinder[39] (v.2.3.8).

## Heterologous expression of candidate genes and protein purification

Plasmids for expression of *bxdA* (N-acyl homoserine lactonase family protein), *bxdD* (aldehyde dehydrogenase family protein), *bxdG* (VOC family protein), and *bxdN* (NAD(P)-dependent oxidoreductase) were ordered from Twist Bioscience. The DNA sequences of the genes were used to generate codon-optimized nucleotide sequences for expression in *E. coli*, applying the default settings. Sequences were introduced to expression plasmid pET28a(+) with *Bam*HI and *Hind*III restriction sites (Twist Bioscience HQ, San Francisco, US). All genes were amplified with Platinum Superfi polymerase II (Thermo Fisher Scientific, Waltham, USA) according to the manufacturer's instructions by using the primers listed in Supplementary Table 2. Then candidate genes were cloned in the expression vector pOPINF (N-terminal His tag) digested with *Hind*III-HF and *Kpn*I-HF. Cloning was performed with In-Fusion (Takara Bio, Shiga, Japan) according to manufacturer protocol and transformed into chemically competent *E. coli* Top10 (NEB, Ipswich, US) and plated on LB plates (25 g/L Luria-Bertani agar, Carl Roth, Karlsruhe, Germany) supplemented with carbenicillin 100 μg/mL (Sigma–Aldrich, St. Louis, USA). Plasmids were isolated from recombinant colonies and the identity of the inserted sequences was confirmed by Sanger sequencing. Next, the constructs were used to transform chemically competent *E. coli* BL21 (DE3) (NEB, Ipswich, US). Correct uptake of the plasmids was verified through colony PCR with vector-specific primers (see above). Positive colonies were inoculated in 5 mL LB with carbenicillin 100 μg/mL and grown overnight at 37 °C, 220 rpm. 100 μL of the pre-culture were inoculated in 100 mL 2xYT media with carbenicillin 100 μg/mL and incubated at 37 °C, 220 rpm until they reached $OD_{600} = 0.5$–0.6. At this point, cultures were incubated for 15 min at 18 °C, 220 rpm and then induced with IPTG 0.5 mM and incubated at 18 °C, 220 rpm for 16 h. For purification, the cultures were harvested by centrifugation at $3200 \times g$, 10 min and resuspended in 10 mL of buffer A1 (50 mM Tris-HCl pH 8, 50 mM glycine, 500 mM sodium chloride, 20 mM imidazole, 5% v/v glycerol, pH 8) supplemented with 0.2 mg/mL Lysozyme and EDTA free protease inhibitor cocktail (cOmplete, Roche, Basel, Switzerland) and incubated for 30 min on ice. Cells were disrupted by sonication using a Sonics Vibra Cell at 40% amplitude, 3 s ON, 2 s OFF, and 2.5 min total time. The crude lysates were centrifuged at $35,000 \times g$ for 30 min and the cleared lysates incubated with 200 μL Ni-NTA agarose beads (Takara Bio, Shiga, Japan) for 1 h at 4 °C. The beads were then sedimented by centrifugation at $1000 \times g$ for 1 min and washed 4 times with buffer A1 before eluting the proteins with buffer B1 (50 mM Tris-HCl pH 8, 50 mM glycine, 500 mM Sodium Chloride, 500 mM imidazole, 5% v/v glycerol, pH 8). Dialysis and buffer exchange were performed using buffer A4 (20 mM HEPES pH 7.5; 150 mM NaCl) in centrifugal concentrators (Amicon Ultra – 10 kDa, Merk Millipore Cork IRL). Proteins were aliquoted in 50 μL and stored at −20 °C. Protein concentration was determined spectrophotometrically at 280 nm on a NanoPhotometer N60 (Implen, Munich, Germany) considering the molecular weight and extinction coefficient. Protein purity and size were checked through SDS-Page on Novex WedgeWell 12% Tris-Glycine Gel (Invitrogen, Waltham, US). The protein ladder used was Colour Protein Standard Broad Range (NEB, Ipswich, US).

## Enzyme assays and product analysis

All reactions were performed in a total volume of 100 μL, in 25 mM potassium phosphate buffer, pH = 7.5 with 5 μg protein. AMPO formation was tested by supplementing the enzyme with 1 mM MBOA or BOA (30 mM stock in MeOH, Sigma–Aldrich, St. Louis, USA). In addition, BxdD was supplemented with NADP+ and BxdN with NADP+ and NADPH. Reactions were initiated by protein addition and incubated at 30 °C, 300 rpm for 2 h in the dark. Reactions were quenched by the addition of 100 μL MeOH, incubated on ice for 15 min and then centrifuged at $15,000 \times g$ for 15 min. The reactions were filtered through 0.22 μm PTFE syringe filters and then transferred to LC–MS glass vials.

LC–MS analysis was performed on a Dionex UltiMate 3000 UHPLC (Thermo Fisher Scientific, Waltham, USA) equipped with Phenomenex Kinetex XB-C18 column (100 × 2.1 mm, 2.6 μm, 100 Å, column temperature 40 °C) coupled to a Bruker Impact II Ultra-High-Resolution Quadrupole-Time-of-Flight mass spectrometer (Bruker Daltonics) equipped with EVOQ Elite electrospray ionization. Analytical conditions consisted of A: $H_2O$ + 0.1% FA and B: ACN, 0.6 mL/min flow with the following gradient: 0–1 min, 15% B, 1–6 min, 15–35% B, 6.1–7.5 min, 100% B, 7.6–10 min, 15% B. Mass-spectrometry data were acquired through ESI with a capillary voltage of 3500 V and end plate offset of 500 V, nebulizer pressure of 2.5 bar with a drying gas flow of 11.0 L/min and a drying temperature of 250 °C. The acquisition was performed at 12 Hz with a mass scan range from 80 to 1000 m/z. For tandem mass-spectrometry ($Ms^2$) collision energy, the stepping option model (from 20 to 50 eV) was used.

## Homology search of BxdA across other bacterial genomes

To investigate how widespread the N-acyl homoserine lactonase enzyme BxdA is, we blasted the amino acid sequence of BxdA in other bacterial genomes. We searched against the microbacteria used in this study (Fig. 3), across the other maize root bacteria strains of the MRB collection and in publicly deposited bacterial genomes. To identify homologues in the microbacteria and the MRB collection, we used the OpenGenomeBrowser[61] using default parameters. To identify homologues to publicly available genomes, we blasted BxdA against the Integrated microbial genomes/microbiome (IMG/M) database run by the Joint Genome Institute (JGI)[40] using default parameters.

## Generation of Δ3bxdA mutant in *Sphingobium* LSP13

A markerless gene deletion mutant lacking the three *bxdA* homologues present in *Sphingobium* LSP13 was produced by sequential in-frame deletion via double homologous recombination using two previously described mutagenesis systems[74,75]. For a list of primers and plasmids used, see Supplementary Tables 3 and 4. Mutagenesis plasmids were prepared using standard molecular cloning protocols[76]. Gene flanking regions containing the bases coding for the first few and the last few amino acids were amplified from genomic DNA and cloned in tandem into pAK405[74] (for MRBLSP13_002921) or into pTETSIX[75] (for MRBLSP13_002227 and MRBLSP13_003006) using a 6 nt restriction linker. Insert sequences were confirmed by Sanger sequencing and final mutagenesis plasmids were transformed into *E. coli* S17-1 (λ/pir)[77]. Mutagenesis plasmids were subsequently transferred to *Sphingobium* LSP13 wild-type or derivative mutants by conjugal transfer[74]. Briefly, the *Sphingobium* acceptors were cultured in 50% LB at 28 °C and the *E. coli* donors in LB supplemented with kanamycin (50 μg/mL) or tetracycline (10 μg/mL) at 37 °C. Donors and acceptors were washed once with 10 mM MgSO4 and resuspended to an $OD_{600}$ of ~30. The suspensions were mixed at a 1:1 ratio and spotted on minimal medium[54] agar without carbon source for 20 h at 28 °C. The cell material was taken up in 10 mM MgSO4 and dilutions plated on media selective for merodiploids. For introduction of the pAK045 derivative pAW02, merodiploids were selected on minimal medium agar containing glucose (20 mM), kanamycin (50 μg/mL) and carbenicillin (50 μg/mL). Individual merodiploid colonies were restreaked once on minimal

medium containing glucose, kanamycin and carbenicillin. Cell material was then resuspended in 10 mM $MgCl_2$, and different dilutions plated onto minimal medium containing glucose, carbenicillin and streptomycin (100 µg/mL) to select for cells having undergone a second crossover event. Resulting colonies were restreaked onto minimal medium containing glucose and either kanamycin or carbenicillin and streptomycin. Those colonies growing only on the latter were further screened, purified and mutation of gene MRBLSP13_002921 confirmed using appropriate primer combinations (Supplementary Table 4). We noticed that with this mutagenesis system, many of the cells growing on carbenicillin and streptomycin showed growth on kanamycin even when the pAK405 plasmid backbone containing the kanamycin resistance cassette was lost after the second homologous recombination. Further mutations were therefore introduced with pTETSIX-derived mutagenesis plasmids (Supplementary Table 3). For transformants with these plasmids, merodiploids were selected on minimal medium supplemented with glucose, carbenicillin and tetracycline (10 µg/mL). Subsequently, merodiploids were subcultured continuously in minimal medium containing glucose without antibiotics. Cells having undergone a second homologous recombination were enriched by sorting non-fluorescent cells on a FACS Sorter Aria III (10.040) as described[75]. Dilutions of the sorted cells were plated on minimal medium containing glucose. Colonies that did not show mCherry fluorescence were further screened, purified and mutation genotype confirmed for all three genes using appropriate primer combinations (Supplementary Table 4).

## Statistical analysis

We used R version 4.0 (R core Team, 2016) for statistical analysis and visualization of the data. All code used for statistical analysis and graphing is available from https://github.com/PMI-Basel/Thoenen_et_al_AMPO_formation. For the analysis of bacterial colonisation, we used log-transformed data. We checked for normality using Shapiro–Wilk-test. Using $t$-test or ANOVA we tested for variance. $P$-values were adjusted for multiple testing using the Benjamini-Hochberg method within R. Raw chromatogram data were peak integrated using MassLynx 4.1 (Waters, Milford, US), using defined properties for the reference compounds in the standards. We used the following packages for data analysis and visualizations: Tidyverse[78], Broom[79], DECIPHER[80], DESeq2[71], emmeans[81], ggthemes[82], pheatmap[83], multcomp[84], phyloseq[85], phytools[86], vegan in combination with custom functions.

## Reporting summary

Further information on research design is available in the Nature Portfolio Reporting Summary linked to this article.

## Data availability

The bacterial genome data (raw reads and annotated genomes) is available from the BioProjects PRJNA1009252 (MRB collection[21]) and PRJNA1083430 (new strains). We have deposited the sequences of the transcriptome experiment at NCBI's Gene Expression Omnibus (GEO) under the accession number GSE263275. We have deposited all source data together with all analysis scripts underlying all figures on GitHub.

## Code availability

All code used for statistical analysis and graphing is available from GitHub.

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

## Acknowledgements
We thank Prof. Julia Vorholt (ETH Zurich) for sharing the pAK405 plasmid and bacterial isolates from the AtSphere leaf collection and Prof. Paul Schulze-Lefert (MPMI Cologne) for providing strains from the AtSphere root and soil collection. Further, we thank Prof. em. Hans-Martin Fischer and Dr. Raphael Ledermann for the plasmid pTETSIX and technical advice. Thanks go to Corinne Suter for support with culturing bacteria and plating assays and to Mirco Hecht for supporting metabolomic analysis. Further, we thank Dr. Thomas Roder for the support with the open genome browser, Dr. Pamela Nicholson from the Next Generation Sequencing Platform in Bern for technical support with sequencing. Additionally, we also thank Dr. Svitlana Malysheva and Stella Stefanova from the Biozentrum FACS Core Facility in Basel for support with FACS analyses. This work was mainly supported by the Interfaculty Research Collaboration "One Health" of the University of Bern. It has also received support by grants of the Austrian Academy of Sciences, the European Union's Horizon 2020 programme (No. 716823 to C.B.), the European Research Council (No. 949595 to C.R.) and the Swiss National Science Foundation (No. 189071 to C.R.). Finally, the work was supported by the State Secretariat for Education, Research and Innovation SERI-funded ERC Consolidator Grant "mifeePs" (No. M822.00079 to K.S.).

## Author contributions
L.T., M.E., and K.S. designed research; L.T. performed microbial plating assays, metabolomic assays, in vitro growth assays, performed the transcriptome experiment, and selected candidate genes. M.K. performed comparative genomics and analysed transcriptomic data. C.P. and A.W. created and tested the *Sphingobium* mutant. M.F. expressed the candidate genes in *E. coli* and tested purified proteins. P.M. performed NMR analyses, P.M., C.A.M.R., T.Z. and E.K. developed and performed metabolomic analyses. C.G. and L.R. tested Arabidopsis isolates for AMPO formation, V.G. cultivated plants, and M.D.N. conducted in vitro growth assays. S.H., C.B., N.S., C.A.M.R., and T.G.K. provided technical infrastructure and R.B. provided new analytic tools. L.T., C.P. and M.K. analysed the data and L.T., C.P., M.E. and K.S. wrote the manuscript. All authors revised the paper.

## Competing interests
The authors declare no competing interests.
