## [Peer Review File · Nature Communications]

The lactonase BxdA mediates metabolic specialisation of maize root bacteria to benzoxazinoidsEditorial Note: Parts of this Peer Review File have been redacted as indicated to maintain the confidentiality of unpublished data.

REVIEWER COMMENTS

Reviewer #1 (Remarks to the Author):

In this study, Thoenen et al discovered a novel lactonase BxdA playing roles in maize root-associated microbial degradation of benzoxazinoids through methylation pathway. The authors aimed to demonstrate that the special function of host-dependent microorganisms (such as MOBA→AMPO metabolism) is an adaptive strategy of microorganisms to plants during long-term evolution. To do so, they assessed the species and abundance of specific species in the maize root microbiome that have the capacity to degrade MBOA and produce AMPO. Then, they compared the number of species that could convert MBOA to AMPO in root microbiota from maize, wheat, oilseed rape and Arabidopsis using a dilution plating method, and found that maize rhizosphere enriched the special species that metabolized MBOA to AMPO due to the accumulation of MBOA around the roots. As a follow-up to this analysis, the authors took *Microbacteria* as the example to reveal that specific maize root microorganisms with the capacity to metabolize MBOA to AMPO contain the conserved gene BxdA encoding N-acyl homoserine lactonase family protein. This is a large body and a good start of work presenting an interesting idea.

My first major concern is that although the author emphasizes the intriguing concept of "metabolic adaptation" in the title, there is insufficient evidence in the main body to prove that the stress of plant metabolites causes directed genetic changes in microbial metabolic levels to adapt to the chemical environment from an ecological and evolutionary perspective. Based on the title, I supposed to see a conserved gene in most of the MBOA-metabolizing taxa in the rhizosphere of maize instead of only one genus. Additionally, I suggest quantifying the abundance of bxdA in the rhizosphere of WT and bx1 to confirm if this gene was enriched in the rhizosphere of WT. The authors tried to use the genus *Microbacteria* from different plants (maize/*Arabidopsis*) to show that the microbial degradation of MBOA into AMPO by maize root microorganisms is based on the lactonase encoded by the BxdA gene. I think it should be compared whether this specific BxdA is present in soil microorganisms or in other plant root microorganisms that can degrade BOA or MBOA. If so, whether it can be compared with its evolutionary conserved differences based on different hosts?

My second major concern is that the text needs a substantial amount of editing (see examples below). It is very difficult to follow the narrative, and experimental result description is too scattered and does not focus on the main topic, and the conclusion given in the preceding paragraphs are inconsistent. The writing and review of the paper is relatively rudimentary. Taking main figure legends as an example, there are a lot of errors and omissions (see details below). The experimental results were described repeatedly, and the main views in the context are inconsistent or even contradictory. I have tried to illustrate some examples below.

Abstract

Line 7-8 “By contrast, bacteria isolated from Arabidopsis, which does not produce benzoxazinoids, were unable to metabolise MBOA.” Here is Arabidopsis does not produce benzoxazinoids, but not bacteria; while that is “bacteria” were unable to metabolise MBOA, not Arabidopsis. Please restate this sentence to make it clearly.

Introduction

The entire introduction needs to be carefully adjusted. The context for the feasibility of metabolic adaptation of plant microbiota is missing currently. However, it is the highlight of your research. I propose to explain the cause and the possible pathways of the development of microbial metabolic adaptation from the perspective of microbial-plant co-evolution. For example, how the selection effect of the rhizosphere environment changes the survival strategy of microbiota, and how to feedback to host plant, etc.

Line 38-48 BOA is a concrete example that support the hypothesis of metabolic adaptation of plant microbiota. Therefore, the general introduction to the BOA could be shortened. Conversely, as to why the metabolism of BOA by soil microbiota does not go through the Methylation pathway, while the plant microbiota does? Is this related to the metabolic adaptation of plant microbiota?

Line 66-69 The two sentences just repeated the same context.

Line 71-74 “Pseudomonas or Sphingobium” are common in both soil and plant. Why you take them as the example of “native bacteria”?

Line 75-87 Here the authors just listed the examples of how soil microbiota degrade benzoxazinoids. However, the ecological evolutionary significance of this metabolism is not fully reflected. Does this metabolism fit the environmental metabolic adaptation hypothesis?

Line 92-97 “metabolic adaptation” is difficult but novel and profound. The authors need to be careful not to write the paper which topic is solely as finding that BxdA is the key to the metabolization of BOA by root microbiota through methylation.

Results

To confirm whether AMPO-forming bacteria are specifically enriched in roots of maize, the authors screened a collection of Arabidopsis bacteria. However, 22 of 57 strains were isolated from leaf of Arabidopsis. Since benzoxazinoids are secreted by roots, why not compare the MBOA-degrading ability of collections of bacteria from the rhizosphere of Arabidopsis and maize.

The authors found that the degradations of DIMBOA-Glc and MBOA benefits bacterial growth, which is not surprising. However, the authors need to determine whether the degradations of DIMBOA-Glc and MBOA or the AMPO-formation benefits maize plants.

Line 107 “to define the distribution of AMPO-bacteria” in what? or identify AMPO-bacteria? In fact, you are looking at the proportion of AMPO-bacteria in the phylogenetic classification of maize culturable root microbial species. After that, it is further classified into three categories (strong, weak, no former). And why do you divide into the three categories? At least it has nothing to do with the content of your subtitle. Is it related to the selection of Microbacteria for gene mining? If yes, you should give clues or foreshadowing.

Line 109 Where are 151 strains from? Any background of them? Are these strains included in the maize root microbiome datasets you mentioned in Line 119?

Line 122 Fig.1A not “Fig.1a”. Change the capitalization of the legend letters throughout the article.

Line 124-127 The legend of Fig.1B is missing. It’s difficult to follow why the pot experiment set with “benzoxazinoid-deficient BX1 mutant” here. The logical flow jumped too much.

Line 157 What is “the classification approach”? Where are the 57 strains from? How do you get them?

Line158-162 You’ve declared that the lowest portion of AMPO-forming on Arabidopsis (-0.002%). I’m confused that why you choose Arabidopsis bacteria to compare maize strong AMPO-formers? Of course, less strong AMPO-formers would be found. Comparatively, Brassica would be better to choose for comparison.

Line 181-185 “For chemical validation of the AMPO-formers and to” If the purpose here is validate chemical AMPO, what is the purpose for results line 102-106? And how you illustrate the credibility of your previous results? Moreover, why do you choose the 50 bacterial isolates? Any information or background of them?

Line 208 change “>30%” to “>30% but <90%”

Line 239 how do you judge AMPO is the “only product”? It is not possible to prove that other substances do not exist (current detection methods do not detect other compounds).

Line 251 This is a typical example that opinions are inconsistent in the different paragraphs. The authors emphasized that the degradation of DIMBOA and MBOA are key steps to form AMPO in this paragraph. However, in line 201-213 “only a few stains subsequently formed low amount of AMPO” (Line 211). Since DIMBOA-Glc degraders and MBOA-degraders are different (uncouple traits) why you conclude that the degradation of both DIMBOA and MBOA are important for AMPO formation?

Line 254-262 I don’t think results from Fig.4D support “AMPO-formation benefit bacterial growth.” First, the data shown lacks statistical analysis. Second, most of isolates (except LBA) grow best in the TSB compared with MBOA and DIMBOA-Glc addition culture. That means, these chemicals more or less inhibit the growth of these microbes. We can judge the chemical tolerance of different microorganisms to the growth instead of microbial growth benefit.

Line 264 This paragraph is very abrupt. Why you have to choose Microbacteria as the subject for this part of study? Why not Sphingobium which are strong AMPO-formers.

Line 267. Four strains of Microbacterium isolated from plants other than maize and Arabidopsis were used, while I only found 3 strains from the Dataset S1. I think one strain isolated from clover is missing. Similarly, information of strain root280d1 of Fig. 4 was also missing from the Dataset S1. Additionally, the authors had more than 40 strains of Microbacterium, how the authors selected the 18 strains from them?

Line 298 What do “approach 17” and “analysis 108” mean?

Line 299-301 I don’t understand why the authors simply selected a single strain LMB2 here for genome analysis? Why don’t include other species of Microbacteria?

Line 344-352 Yes, the author found that many strong AMPO-formers in maize root contain *bxdA*, but what about other strong AMPO-formers from other plant species? For example, strong AMPO-formers from wheat and oilseed rapeseed? Do they have the same or similar *bxdA* genes? That is important to support “metabolic adaptation”.

Figures

Fig. 1 No legend in Fig.1B. WT Replicates in Fig 1C are different: Root=8; rhizosphere=10; Soil=9
Why?

Fig. 3D the ordinate must be “bacterial growth” instead of “AUC”. No statistical analysis for different treatments.

Fig.4 “AMPO former” and “non-AMPO former” What are the boundaries of the evaluation? Is there a threshold for the measurement?

I don’t understand the heatmap of the row KHB019: blank squares with short black dashes inside.
What does it mean?

Line 291 change “seven to eleven” to eight to twelve

Fig.S5 “DIMBOA-Glc” is missing in the abscissa of DIMBOA-Glc tiles in Fig.S5C.

Figures of S5D and S5F is missing with legends listed below.

Hope these suggestions could improve the manuscript. Good luck!

Reviewer #2 (Remarks to the Author):

The manuscript by Thoenen et al. explores the mechanisms of bacterial adaptation to maize roots that secrete benzoxazinoids. The authors previously isolated bacteria from maize roots and compared their benzoxazinoid tolerance with those from *Arabidopsis* roots. Their study was recently published in PNAS. This research serves as a follow-up to their previous report, where the authors initially probed the prevalence of MBOA metabolizing bacteria in maize roots. Utilizing AMPO-forming activity, the authors observed a higher abundance of AMPO-forming bacteria in

maize roots compared to those isolated from Triticum, Medicago, Brassica, and Arabidopsis roots. AMPO-forming bacteria exhibit phylogenetic diversity, and several members can utilize benzoxazinoids as a solo carbon source. To delve into the molecular mechanisms of benzoxazinoid metabolism, the authors focused on Microbacteria, isolating both AMPO-forming and non-AMPO-forming members. Through genomic and transcriptomic analyses, the authors identified a gene cluster for benzoxazinoid metabolism, characterized as the Bxd cluster, having 15 genes encoding 13 enzymes and two transcriptional regulators. Among these, the gene responsible for the initial step of MBOA metabolism, BxdA, was identified using a recombinant protein expressed in E. coli. This study analyzed the molecular mechanisms of plant-microbiota interactions facilitated by plant-specialized metabolites, though several issues need addressing before concluding that bxdA mediates bacterial adaptation to maize roots.

Major comments

1. While the authors characterized the in vitro function of bxdA in benzoxazinoid metabolism, in vivo characterization was not demonstrated. It would be highly recommended to generate a bacterial mutant and investigate its adaptation to maize roots. While the study states that obtaining a mutant in Microbacteria is technically challenging, it could be performed using other MBOA-metabolizing bacteria like Sphingobium or Pseudarthrobacter. LSP13 contains a homologous gene and could serve as a suitable strain for generating gene disruption mutants. As Sphingomonadaceae comprises abundant members in maize roots and Sphingobium is the most prevalent AMPO-former, analyzing interactions in both Microbacteriaceae and Sphingomonadaceae will significantly enhance our understanding of these interactions.

2. The Bxd gene cluster encompasses 13 enzyme genes, yet most were challenging to characterize in vitro due to difficulties in protein expression in E. coli. Has the author explored other heterologous expression systems? Additionally, there is a lack of information on the substrate specificity of BxdA. Is it capable of metabolizing other lactones present in the rhizosphere?

3. Maize root secretes DIMBOA-Glu, which is converted to AMPO. The authors highlight the common occurrence of the deglycosylation step in soil bacteria, and certain root-colonizing members can use MBOA as a carbon source, resulting in the formation of AMPO. In soil, AMPO is further metabolized to produce 2-acetylamino-9-methoxy-2-amino-3H-phenoxazin-3-one (AAMPO). Does the maize rhizosphere harbor bacteria that metabolize AMPO? Additionally, are AMPO levels higher than AAMPO in the rhizosphere soil of maize? Do the rhizosphere soil of maize contain more AMPO than AAMPO? It would be beneficial for readers if the authors discussed their findings within the framework of metabolite profiles present in the maize rhizosphere soil.

Minor comments

1. Fig.3. Carbons released in the catabolism from MBOA to AMP would be considerably less compared to those from DIMBOA-Glu or glucose at the same concentration. I would recommend the authors to show the growth curve of the bacterial isolates in Supplementary Figures not just AUC for easier comparison of growth based on the different substrates..

2. There are both AMPO-forming and non AMPO-forming members in the Family. This aspect should be explained in Fig S4 as well.

3. Fig. S8E. What does “gene” in X axis mean?

Reviewer #3 (Remarks to the Author):

In the current study, the authors aimed to address the mechanisms by which root-associated bacteria metabolize root-secreted benzoxiazinoids. Using the accumulation of the red color-producing metabolite, AMPO, the authors identified a series of bacterial strains that are able to accumulate AMPO when treated with MBOA, one of the abundant benzoxiazinoids secreted from maize roots. This capacity was enriched in maize-derived isolates compared to the isolated from other plant species and exerted a growth fitness in MBOA-catalyzing isolates, implying that this is an adaptive feature of maize-associated microbiota. Then, by employing integrative bioinformatic approaches, including comparative genomics and transcriptomics, they identified the *bxd* cluster that possibly mediates MBOA catabolism. Although its *in vivo* function remains to be seen, one of the gene products, *bxdA*, was shown to possess MBOA-catalyzing activity *in vitro*, supporting this hypothesis. Overall, this study revealed a molecular framework that might have allowed maize plants to recruit their specific structure of root microbiota. The manuscript is well-written and well-structured, and the details are paid careful attention. To what extent this system contributes to maize fitness, for example, by exerting fitness effect, etc, remains to be seen but is an intriguing open question.

An important question that remains to be addressed is how this capacity has been acquired in the studied Microbacteria lineage. It is almost certain that this gene cluster was horizontally acquired, and the authors also showed that there are some other bacterial strains that have *BxdA* homologs. The authors concluded that *BxdA* was rarely found in other bacteria, but I do not see this, as Figure S9B clearly identified genes with high (>60%) similarity with *BxdA*, which ranged from Actinobacteria to Beta/Gamma-proteobacteria. It was also unclear why the authors focused on the top 30 hits. Also, the comparison within Microbacteria lineage was limited to maize and *Arabidopsis* isolates, and many other strains were not taken into account. Overall, the analysis of *BxdA* gene distribution is limited to a biased genome dataset and does not support the authors'

claim. To claim that it is a specific innovation in the MBOA-catabolizing isolates associated with the roots of benzoxazinoid-producing plants (lines 414-415), it is necessary to perform a more comprehensive and unbiased genomic search for BxdA using a custom set of publicly available genome sequences.

I am also curious to know how the authors put this finding into the microbial community context. Apparently, not all bacterial strains catabolize DIMBOA-Glc down to AMPO but only until MBOA, while some bacteria catabolize only MBOA but not DIMBOA-Glc. These findings, depicted in lines 207-213, clearly suggest metabolic complementation between root microbiota. In this context, I would be curious to know whether those only catabolizing DIMBOA-Glc and those only catabolizing MBOA would co-occur in natural communities, in which case the community-context metabolic complementation can be clearly suggested. Also, in this context, I was wondering if AMPO produced by these root bacteria can be further utilized by other bacteria and, alternatively, if AMPO is abundantly detected in maize rhizosphere without further catabolism.

Below are some minor comments:

- Something is wrong with the Fig 1 legend. Please revise it to fit with its content.
- How do the authors explain the difference in AMPO production of Arabidopsis isolates between plate- and liquid-based screening?

Point-by-point response letter

Maize root bacteria degrade host-specialized metabolites through the lactonase BxdA

By Lisa Thoenen et al.

We appreciate the generally positive perception of our work by the editor and the reviewers. We have addressed all major and minor concerns and incorporated them in the revised version of our manuscript. Overall, we are grateful for the critical and rigorous assessment of our manuscript and we see that it has substantially improved with the reviewer's constructive feedback. For transparency, we provide the revised manuscript in a 'clean' version and in a version that highlights the 'changes' compared to the original submission using track/change. Here we respond to the reviewers' comments on our manuscript in a detailed point-by-point manner. Our responses are colored in blue. The line numbers mentioned refer to the revised main text document without tracked changes.

REVIEWER #1 (REMARKS TO THE AUTHOR):

In this study, Thoenen et al discovered a novel lactonase BxdA playing roles in maize root-associate microbial degradation of benzoxazinoids through methylation pathway. The authors aimed to demonstrate that the special function of host-dependent microorganisms (such as MBOA→AMPO metabolism) is an adaptive strategy of microorganisms to plants during long-term evolution. To do so, they assessed the species and abundance of specific species in the maize root microbiome that have the capacity to degrade MBOA and produce AMPO. Then, they compared the number of species that could convert MBOA to AMPO in root microbiota from maize, wheat, oilseed rape and Arabidopsis using a dilution plating method, and found that maize rhizosphere enriched the special species that metabolized MBOA to AMPO due to the accumulation of MBOA around the roots. As a follow-up to this analysis, the authors took *Microbacteria* as the example to reveal that specific maize root microorganisms with the capacity to metabolize MBOA to AMPO contain the conserved gene BxdA encoding N-acyl homoserine lactonase family protein. This is a large body and a good start of work presenting an interesting idea.

We noticed upon reading of the comments that there were potential confusions in our manuscript text. We have tried to improve this by substantial editing. One such example of a confusions is the 'methylation pathway'. In our manuscript we do not refer to a methylation pathway as no methylation is involved in the topic of this study. Reviewer #1 may refer to the distinction between methoxylated and non-methoxylated benzoxazinoid compounds, which are chemical variants originating from the same biosynthesis pathway how plants produce benzoxazinoid compounds (see detailed explanation below).

My first major concern is that although the author emphasizes the intriguing concept of "metabolic adaptation" in the title, there is insufficient evidence in the main body to prove that the stress of plant metabolites causes directed genetic changes in microbial metabolic levels to adapt to the chemical environment from an ecological and evolutionary perspective. Based on the title, I supposed to see a conserved gene in most of the MBOA-metabolizing taxa in the rhizosphere of maize instead of only one genus. Additionally, i suggest quantifying the abundance of bxdA in the rhizosphere of WT and bx1 to confirm if this gene was enriched in the rhizosphere of WT. The authors tried to use the genus *Microbacteria* from different plants (maize/*Arabidopsis*) to show that the microbial degradation of MBOA into

AMPO by maize root microorganisms is based on the lactonase encoded by the BxdA gene. I think it should be compared whether this specific BxdA is present in soil microorganisms or in other plant root microorganisms that can degrade BOA or MBOA. If so, whether it can be compared with its evolutionary conserved differences based on different hosts?

From the reviewer's comment, we have realized that the framing of our work with “metabolic adaptation’ can be misinterpreted. Our experimental work was not designed to answer ecological or evolutionary questions related to bxdA. Therefore, we have toned down the claim of “metabolic adaptation” and also removed it from the title of the manuscript.

The reviewer states that for metabolic adaptation s/he would expect a “conserved gene in most of the MBOA-metabolizing taxa in the rhizosphere of maize instead of only one genus”. This is actually the case. Consistent with the MBOA degradation and AMPO forming phenotypes, we found *bxdA* homologues with high protein similarity in *Pseudoarthrobacter* LMD1 and *Sphingobium* LSP13, LMA1 and LMC3. We rephrased the results section to make clear that *bxdA* is also present in genomes of other AMPO forming maize root bacteria (See Line 371...). Also, the new data on the Δ 3*bxdA* mutant in *Sphingobium* will help to avoid such a misinterpretation.

The reviewer also asked to compare whether BxdA is present in other soil or plant root bacteria. A similar question was raised by Reviewer#3. We have performed a new homology search by blasting BxdA of LMB2 against a large set of publicly available genomes of the Integrated Microbial Genomes/Microbiome (IMG/M) database of the Joint Genome Institute. Out of >123K genomes from isolates only 28 genomes had *bxdA* genes with similarities higher >60% (amino acid similarity). Interestingly, these highly similar BxdA hits were found only in few bacterial taxa including Microbacteriaceae, Sphingomonadaceae, Micrococcaceae, Burkholderiaceae and Pseudomonadaceae, most of which we had identified and confirmed to form AMPO with our isolate collection approach. Unfortunately, the incomplete metadata of the IMG/M genomes precluded a systematic analysis of compartment (soil vs. root) and host origin (plant species). These findings are reported in the new Figure S12C.

Reviewer #1, additionally suggested to quantify the abundance of BxdA in the rhizosphere of WT and *bx1* to confirm if this gene was enriched in the rhizosphere of WT. We have inspected our rhizosphere metagenome data of WT and *bx1* mutant plants. For this analysis, we mapped the *bxdA* genes from genomes of strains that form AMPO to the metagenomic reads and calculated the mapping depth in each sample. The mapping depth was adjusted by library size to normalize for differences in sequencing depths between the different samples. The Figure below reports the mapping depths of each gene in the WT and *bx1* rhizosphere samples (n=4). This analysis reveals a trend that generally more reads, to which BxdA genes map, are found in WT compared to *bx1* rhizospheres. It also shows that the three gene copies of the *Sphingobium* strains LMA1, LMC3 LSP13 are more prevalent compared to the gene copies of the various microbacteria. This finding reflects the higher abundance of *Sphingobium* on roots compared to microbacteria as determined by amplicon sequencing (Fig. 1d). Although this data points to an enrichment of *bxdA* in WT maize rhizospheres, we think that for the question of metabolic adaptation the critical comparisons with other plant hosts are missing. *BxdA* gene quantifications should compare maize with other plant species, e.g. analogous to root extract plating of wheat, Medicago, Brassica and Arabidopsis (Fig. 1c). However, generating rhizosphere metagenomes of other plants grown in the same soil would be beyond reasonable scales. We show this data in response to Reviewer #1, but we do not see an added value to include it in this study.

[Redacted]

My second major concern is that the text needs a substantial amount of editing (see examples below). It is very difficult to follow the narrative, and experimental result description is too scattered and does not focus on the main topic, and the conclusion given in the preceding paragraphs are inconsistent. The writing and review of the paper is relatively rudimentary. Taking main figure legends as an example, there are a lot of errors and omissions (see details below). The experimental results were described repeatedly, and the main views in the context are inconsistent or even contradictory. I have tried to illustrate some examples below. We agree with the reviewer and have substantially revised the manuscript to simplify the narrative. We have reordered experiments to have a straight focus on the main topic. The story line now starts with plating of root extracts that reveals that AMPO formation is characteristic for maize bacteria. Then the single-strain work of the MRB collection allows to conclude that AMPO formation is present in several taxa among abundant maize root bacteria. Third, we report the chemical analyses of the liquid cultures, which uncovered that *strong* MBOA-degraders were strong AMPO-formers. Next – to identify the genetic basis using Microbacteria – we describe the strains’ chemical and growth phenotypes, followed by comparative genomics which led to the identification of the *bx*d gene cluster. Ultimately, we conclude the story line with first *in vitro* confirmation and second *in vivo* validations of BxdA.

In case the reviewer still thinks the story flow is scattered, we see a further possibility to deconvolute the narrative: The data of the DIMBOA-Glc experiments could be removed (Fig. 2b and parts in Figs. 2d, 3 and 4). The data of the strain exposures to DIMBOA-Glc is important for the bigger picture, but not essential for the discovery of the *bx*d gene cluster and *bx*dA.

Abstract: Line 7-8 “By contrast, bacteria isolated from Arabidopsis, which does not produce benzoxazinoids, were unable to metabolise MBOA.” Here is Arabidopsis does not produce benzoxazinoids, but not bacteria; while that is “bacteria” were unable to metabolise MBOA, not Arabidopsis. Please restate this sentence to make it clearly.

We thank the reviewer to identify this linguistic mistake. With the rewritten abstract, this change became unnecessary.

Introduction: The entire introduction needs to be carefully adjusted. The context for the feasibility of metabolic adaptation of plant microbiota is missing currently. However, it is the highlight of your research. I propose to explain the cause and the possible pathways of the development of microbial metabolic adaptation from the perspective of microbial-plant co-evolution. For example, how the selection effect of the rhizosphere environment changes the survival strategy of microbiota, and how to feedback to host plant, etc.

We have refined the context and explain now possible advantages for bacteria when adapting to the metabolites of their host plant. We did this from the perspective of the bacteria only, because our experimental design and results do not allow to conclude on microbial-plant co-evolution (See Line 102...).

Line 38-48 BOA is a concrete example that support the hypothesis of metabolic adaptation of plant microbiota. Therefore, the general introduction to the BOA could be shortened. Conversely, as to why the metabolism of BOA by soil microbiota does not go through the Methylation pathway, while the plant microbiota does? Is this related to the metabolic adaptation of plant microbiota?

We have difficulties to distil the actual criticism of this comment and suspect a misinterpretation of the topical background of the study. There is no ‘methylation pathway’ that should distinguish the soil and plant microbiota. However, there is the distinction of methoxylated and non-methoxylated benzoxazinoid compounds. For instance, BOA is the non-methoxylated sister compound of MBOA (see Fig. S1). Whether benzoxazinoids are methoxylated or not is decided by the specific biosynthesis pathways of the host plant and this does not depend on the soil or plant microbiota. While maize primarily synthesizes methoxylated BXs (DIMBOA-Glc > DIMBOA > MBOA > AMPO), rye or barley mainly produce the non-methoxylated analogues following the same chemical conversion pathway (DIBOA-Glc > DIBOA > BOA > APO). To clarify, we have added to this paragraph: “In maize, the methoxylated benzoxazinoids dominate, while rye only produces non-methoxylated benzoxazinoids, and wheat forms a mixture of both” (See Line 81...).

Line 66-69 The two sentences just repeated the same context.

We thank the reviewer for the critical reading. We merged the two sentences into one.

Line 71-74 “*Pseudomonas* or *Sphingobium*” are common in both soil and plant. Why you take them as the example of “native bacteria”?

To address this unclarity, we rewrote the sentences to explain the origins of the strains better: “For example, *Pseudomonas* isolated from *Arabidopsis* uses triterpenes as carbon sources. *Sphingobium* bacteria isolated from tomato grow on tomatine as a sole carbon source.”

Line 75-87 Here the authors just listed the examples of how soil microbiota degrade benzoxazinoids. However, the ecological evolutionary significance of this metabolism is not fully reflected. Does this metabolism fit the environmental metabolic adaptation hypothesis?

Reviewer #1 identified correctly the distinction between our work compared to previous work. This is actually, why this paragraph ends with the conclusion that “Microbial metabolisation of benzoxazinoids ... not yet been investigated in the native context of root microbes from benzoxazinoid-exuding plants.” Previous work had tested various microbes for benzoxazinoid degradation without reference to their host of isolation. These microbes were isolated mostly from soil and some from non-benzoxazinoid plants. However, the examples with tomatine and triterpenes (the above paragraph) suggest that the ability to

metabolise benzoxazinoids is specific to maize root bacteria. We made the transition to the upper paragraph more explicit (See Line 111).

Line92-97 “metabolic adaptation” is difficult but novel and profound. The authors need to be careful not to write the paper which topic is solely as finding that BxdA is the key to the metabolization of BOA by root microbiota through methylation.

Although the plating of root extracts from different plant species showed that the capacity to form AMPO is limited to maize root microbes (Fig. 1d), we agree, we did not demonstrate that this was through BxdA. Therefore, we have toned down the overall claim of “metabolic adaptation” in the manuscript and rephrased this paragraph (See Line 129...).

Results

To confirm whether AMPO-forming bacteria are specifically enriched in roots of maize, the authors screened a collection of Arabidopsis bacteria. However, 22 of 57 strains were isolated from leaf of Arabidopsis. Since benzoxazinoids are secreted by roots, why not compare the MBOA-degrading ability of collections of bacteria from the rhizosphere of Arabidopsis and maize.

We agree with the reviewer that the screened Arabidopsis strains with the many leaf isolates was not ideal and therefore, does not support the initial claim of “metabolic adaptation”. The reviewer correctly states that this would require the screening of Arabidopsis root bacteria. Our initial goal was not a systematic comparison of the two collections. Instead, we had selected the AtSphere strains with the highest 16S rRNA gene sequence similarity to the strains of the MRB collection. This resulted in 57 strains that were isolated either from Arabidopsis roots or leaves, or the soil where Arabidopsis was grown to build the AtSphere collection. Our rationale was to expand the number of strains and their paired information of genomes and AMPO formation phenotypes for subsequent comparative genomics. Important for successful comparative genomics was that none of the tested microbacteria from Arabidopsis could form AMPO (Fig. S7, confirmed in Fig. S8), but many from maize did (Fig. 1d). We have corrected the context of the screening of the Arabidopsis strains and we now present the results in the supplement (See Line 252..., Fig. S7). Importantly, we do not interpret the data any more related to the hypothesis of metabolic adaptation. We only stress the conclusion that none of the tested Microbacteria from Arabidopsis could form AMPO, which was crucial for comparative genomics (See Line 255...).

The authors found that the degradations of DIMBOA-Glc and MBOA benefits bacterial growth, which is not surprising. However, the authors need to determine whether the degradations of DIMBOA-Glc and MBOA or the AMPO-formation benefits maize plants.

We do not think that re-determining the benefits to maize plants is needed. First, because the chemical ecology of benzoxazinoids has been extensively studied and second, many benefits to maize plants have been shown in the past. We mention them in the introduction (See Line 75...). Host plant benefits include the degradation products of benzoxazinoids, where the allelopathic activities of aminophenoxazinones APO and AMPO have been repeatedly demonstrated (e.g., Niemeyer, *Phytochemistry*, 1988; Schulz et al., *J Chem Ecol*, 2013; Hussain et al., *Environ Exp Bot*, 2022). While benzoxazinoid-exuding host plants like maize tolerate the aminophenoxazinones, their phytotoxic activities result in inhibiting the germination and growth of neighbouring plants. We explain in the discussion (see Line 481...) the logic that host plants like maize benefit from recruiting A(M)PO-forming bacteria as they convert (M)BOA to strongly allelopathic compounds that suppress then competing weeds, which improves host fitness.

At the same time, we agree with the reviewer that the degradations of DIMBOA-Glc and MBOA benefitting bacterial growth are a priori not surprising – but this was not known before and needed to be shown for a first time.

Line 107 “to define the distribution of AMPO-bacteria” in what? or identify AMPO-bacteria? In fact, you are looking at the proportion of AMPO-bacteria in the phylogenetic classification of maize culturable root microbial species. After that, it is further classified into three categories (strong, weak, no former). And why do you divide into the three categories? At least it has nothing to do with the content of your subtitle. Is it related to the selection of Microbacteria for gene mining? If yes, you should give clues or foreshadowing.

We have rephrased this misleading introductory sentence. It now reads “To test how widespread AMPO formation is among maize root bacteria,...” (Line 166...).

We classified the phenotypes in three categories to cover not only the two extreme cases (nothing happened (*non* AMPO-formers) and AMPO was clearly formed (*strong*)), but we also wanted to reflect on the observation that some bacteria like the *Agrobacterium* do something with MBOA (plates turn orangish) without clearly forming red AMPO (see Fig. S3). We think that the classification with the intermediate category (*weak*) is important as it allows the possibility of MBOA metabolizations other than only forming red AMPO. Of note, for all downstream work to identify the genetic basis of AMPO formation, we only focused on the extreme phenotypes (*non* vs. *strong* AMPO-formers), which we mention in the results text (See Line 196...).

Line 109 Where are 151 strains from? Any background of them? Are these strains included in the maize root microbiome datasets you mentioned in Line 119?

We now specify the origin (MRB collection and reference) and background of these strains in the revised version of the manuscript (See Line 166...).

Line 122 Fig.1A not “Fig.1a”. Change the capitalization of the legend letters throughout the article.

We did not capitalize the legend letters in the revised version because other Nature Communications articles use lowercase indexes (e.g., Mesny et al. 2021 or Emmenegger et al. 2023). For consistency, we now adjusted the capitalization in the Figures themselves to match the legends.

Line 124-127 The legend of Fig.1B is missing. It’s difficult to follow why the pot experiment set with a “benzoxazinoid-deficient BX1 mutant” here. The logical flow jumped too much.

We added the missing information for this panel to the legend and adapted the logical flow.

Line 157 What is “the classification approach”? Where are the 57 strains from? How do you get them?

In the revised version, we now explain carefully the origin of the strains and that they were screened for AMPO formation by growing them on MBOA-containing agar plates, i.e. the same assay as for screening the maize strains (See Line 252... and Supplementary Results). We also provide further details on the methods of how they were selected (See Line 574...).

Line158-162 You’ve declared that the lowest portion of AMPO-forming on Arabidopsis (-0.002%). I’m confused that why you choose Arabidopsis bacteria to compare maize strong AMPO-formers? Of course, less strong AMPO-formers would be found. Comparatively, Brassica would be better to choose for comparison.

The reason to work with *Arabidopsis* bacteria is that with the AtSphere collection (Bai et al. 2015, Nature) strains with sequenced genomes were available. So, for comparative genomics we could make use of the publicly available genomes and only needed to phenotype the strains for AMPO formation. *Brassica* strains would have most likely also worked for our purpose, but no collection and no genomes were available at this time. To help with the story flow and the understanding of our rationale, we have now moved the AtSphere collection to the paragraph for selection of strains for comparative genomics.

Line 181-185 “For chemical validation of the AMPO-formers and to” If the purpose here is validate chemical AMPO, what is the purpose for results line 102-106? And how you illustrate the credibility of your previous results? Moreover, why do you choose the 50 bacterial isolates? Any information or background of them?

With the plate assays we grew the strains on MBOA containing plates and we had found that AMPO-formation was reflected with the colour change to red. With UPLC-MS and NMR (former lines 102-106) we confirmed that the red colour is indeed AMPO. Furthermore, the assay with MBOA plates did not allow us to conclude whether maize root bacteria can also degrade MBOA without forming AMPO. Therefore, we did the assay in liquid culture containing MBOA with the goal to validate AMPO formation of each strain in another assay and at the same time to quantify bacterial MBOA metabolism by UPLC-MS. The 46 strains were selected to be a taxonomically broad set of strains representing the MRB collection. We have added this rationale for selection of these strains to the manuscript text (See Line 192...).

Line 208 change “>30%” to “>30% but <90%”
Done (See Line 649...).

Line 239 how do you judge AMPO is the “only product”? It is not possible to prove that other substances do not exist (current detection methods do not detect other compounds).

We rephrased this summary statement that there must be multiple ways for MBOA degradation also with other products than AMPO formed (See Line 216...).

Line 251 This is a typical example that opinions are inconsistent in the different paragraphs. The authors emphasized that the degradation of DIMBOA and MBOA are key steps to form AMPO in this paragraph. However, in line 201-213 “only a few stains subsequently formed low amount of AMPO” (Line 211). Since DIMBOA-Glc degraders and MBOA-degraders are different (uncouple traits) why you conclude that the degradation of both DIMBOA and MBOA are important for AMPO formation?

Thanks for raising this unclarity. It was not our intention to refer to a coupling of degradation traits for DIMBOA-Glc and MBOA degrading strains. We have rephrased these sentences more clearly that the aim was to test strains that degrade either DIMBOA-Glc and/or MBOA degrading strains and whether they can use these compounds as the sole carbon source and benefit in growth (See Line 236...).

Line 254-262 I don’t think results from Fig.4D (3D) support “AMPO-formation benefit bacterial growth.” First, the data shown lacks statistical analysis. Second, most of isolates (except LBA) grow best in the TSB compared with MBOA and DIMBOA-Glc addition culture. That means, these chemicals more or less inhibit the growth of these microbes. We can judge the chemical tolerance of different microorganisms to the growth instead of microbial growth benefit.

We have added the results of statistical analyses to the Fig. 2d. We also replaced the positive control of TSB with the controls of minimal medium supplemented with glucose at the same concentrations (500 and 2'500 μ M). Because TSB contains much more nutrients than the minimal medium we supplied, a direct comparison of growth between TSB and the other treatments is not appropriate. Furthermore, we realized that the TSB control could easily have led to confusion and could have suggested that the other conditions were TSB supplemented with MBOA and DIMBOA-Glc rather than minimal medium supplemented with these carbon sources.

The new control allows molar comparison (which was not possible with the TSB control) for the tested compounds and better judgment of eventual inference of chemical tolerance. The reviewer correctly stated the need to tolerate DIMBOA-Glc and/or MBOA for growth. The growth data is nicely consistent with the earlier tolerance screening. The strong MBOA- and DIMBOA-Glc degraders LSP13, LMD1, LMB2 were also highly tolerant strains and they increased their cell numbers in presence of both MBOA or DIMBOA-Glc (Fig. 2d). The other strains, which were not very tolerant, didn't grow on the provided MBOA concentrations but partially benefitted at higher DIMBOA-Glc concentration possibly making use of the glucose moiety. We have added this consistency to their tolerance levels. See revised paragraphs starting at line 236.

Line 264 This paragraph is very abrupt. Why you have to choose *Microbacteria* as the subject for this part of study? Why not *Sphingobium* which are strong AMPO-formers.

We rewrote the start of this paragraph and give now a detailed explanation why we selected the genus *Microbacterium* to study the genetic basis of AMPO formation (See Line 251...).

Line 267. Four strains of *Microbacterium* isolated from plants other than maize and *Arabidopsis* were used, while I only found 3 strains from the Dataset S1. I think one strain isolated from clover is missing. Similarly, information of strain root280d1 of Fig. 4 was also missing from the Dataset S1. Additionally, the authors had more than 40 strains of *Microbacterium*, how the authors selected the 18 strains from them?

We thank the reviewer for identifying the missing information on the clover strain KHB019. We added this information to the Dataset S1 and also uploaded the genome of KHB019 to NCBI. Although the information for Root280D1 was already in the Dataset S1, we needed to correct the strain ID from Root280 to Root280D1.

Although the MRB collection has a total of 42 *Microbacterium* strains, we only sequenced genomes for strains that differed in their full-length 16S rRNA gene sequence (only one strain was sequenced among strains with identical full-length 16S rRNA gene sequence; see Thoenen et al. 2023, PNAS). We have added this explanation for the selection of the 18 *Microbacterium* strains from maize to the revised manuscript (See Line 801...).

Line 298 What do “approach 17” and “analysis 108” mean?

These numbers refer to the number of genes that were identified by the different approaches. We now phrase it explicitly with “... the kmer approach resulted in 17 candidate genes and the transcriptome analysis found 108 candidate genes.” (See Line 293...).

Line 299-301 I don't understand why the authors simply selected a single strain LMB2 here for genome analysis? Why don't include other species of *Microbacteria*?

For clarification: We did not perform genome analyses only in the single strain LMB2 but the Orthogroup and kmer approaches were performed across all 39 *Microbacterium* genomes. However, we needed a single genome to recapitulate all the identified candidate genes. Because we had performed the transcriptome analysis following MBOA exposure with

LMB2, we selected LMB2 as the reference genome. Moreover, LMB2 is a strong AMPO-former with a simple AMPO chemotype, i.e. forming only AMPO and no other degradation products (Fig. 2c). We added this clarification starting at line 298.

Line 344-352 Yes, the author found that many strong AMPO-formers in maize root contain bxdA, but what about other strong AMPO-formers from other plant species? For example, strong AMPO-formers from wheat and oilseed rapeseed? Do they have the same or similar bxdA genes? That is important to support “metabolic adaptation”.

In the initial submission, the details on the homology searches were hidden in the supplementary results. Because these details matter, we have moved them in the revised version to the main text. Comparing the BxdA protein homologous of all confirmed AMPO-forming strains, we found that they all have amino acid sequence similarities of ~60% or higher. Hence, we could set a threshold of >60% amino acid sequence homology for BxdA homologs to be associated with an AMPO formation phenotype. Further homology searches were then assessed with this similarity threshold. Still, because we do not have further validated AMPO-formers or non-AMPO formers of other plant species, we have toned down the overall claim of metabolic adaptation.

Figures

Fig. 1 No legend in Fig.1B. WT Replicates in Fig 1C are different: Root=8; rhizosphere=10; Soil=9 Why?

The figure legend for 1b has now been added. We initially had 10 replicates for all plants but due to technical problems lost a few samples during sample preparation and this resulted in unequal sample numbers. These unequal sample numbers are acknowledged in the caption.

Fig. 3D the ordinate must be “bacterial growth” instead of “AUC”. No statistical analysis for different treatments.

Done. The y-axis label was changed from AUC to “Bacterial growth [AUC]” and the results of the statistical analysis were added to the figure.

Fig.4 “AMPO former” and “non-AMPO former” What are the boundaries of the evaluation? Is there a threshold for the measurement?

The classifications refer to the non- and strong AMPO-formers as defined on MBOA-containing plates. With this plate assay, we had screened the MRB strains (Fig. 1d) as well as selected strains of the AtSphere collection (Fig. S7). We described this visual evaluation method together with illustrations for the boundaries at the first occurrence in the manuscript text (Line 168..., Fig. S3). It is also described in the methods (Line 588...). We now also mention the classification method in the caption of the figure.

I don't understand the heatmap of the row KHB019: blank squares with short black dashes inside. What does it mean?

We used the visual classifications of the assay on MBOA-containing plates for comparative genomics. For comparative genomics, we included the strain KHB019 with its phenotype on plates (a non-AMPO former) and genome. However, the strain was then not tested in the liquid assays, which we used to confirm the AMPO phenotype of the plate assay. We updated the figure to explain that KHB019 was not tested in the liquid assays.

Line 291 change “seven to eleven” to eight to twelve

Done.

Fig.S5 “DIMBOA-Glc” is missing in the abscissa of DIMBOA-Glc tiles in Fig.S5C. Figures of S5D and S5F is missing with legends listed below.
Done.

REVIEWER #2 (REMARKS TO THE AUTHOR):

The manuscript by Thoenen et al. explores the mechanisms of bacterial adaptation to maize roots that secrete benzoxazinoids. The authors previously isolated bacteria from maize roots and compared their benzoxazinoid tolerance with those from Arabidopsis roots. Their study was recently published in PNAS. This research serves as a follow-up to their previous report, where the authors initially probed the prevalence of MBOA metabolizing bacteria in maize roots. Utilizing AMPO-forming activity, the authors observed a higher abundance of AMPO-forming bacteria in maize roots compared to those isolated from Triticum, Medicago, Brassica, and Arabidopsis roots. AMPO-forming bacteria exhibit phylogenetic diversity, and several members can utilize benzoxazinoids as a solo carbon source. To delve into the molecular mechanisms of benzoxazinoid metabolism, the authors focused on Microbacteria, isolating both AMPO-forming and non-AMPO-forming members. Through genomic and transcriptomic analyses, the authors identified a gene cluster for benzoxazinoid metabolism, characterized as the Bxd cluster, having 15 genes encoding 13 enzymes and two transcriptional regulators. Among these, the gene responsible for the initial step of MBOA metabolism, BxdA, was identified using a recombinant protein expressed in *E. coli*. This study analyzed the molecular mechanisms of plant-microbiota interactions facilitated by plant-specialized metabolites, though several issues need addressing before concluding that bxdA mediates bacterial adaptation to maize roots.

Major comments

1. While the authors characterized the *in vitro* function of bxdA in benzoxazinoid metabolism, *in vivo* characterization was not demonstrated. It would be highly recommended to generate a bacterial mutant and investigate its adaptation to maize roots. While the study states that obtaining a mutant in Microbacteria is technically challenging, it could be performed using other MBOA-metabolizing bacteria like *Sphingobium* or *Pseudarthrobacter*. LSP13 contains a homologous gene and could serve as a suitable strain for generating gene disruption mutants. As Sphingomonadaceae comprises abundant members in maize roots and *Sphingobium* is the most prevalent AMPO-former, analyzing interactions in both Microbacteriaceae and Sphingomonadaceae will significantly enhance our understanding of these interactions.

This is an excellent comment and indeed, *in vivo* confirmation for the function of BxdA was missing in the previous version of the manuscript. We had been working on this issue, which turned out to be not trivial because *Sphingobium* LSP13 has 3 gene copies of homologous *bxdA* genes (MRBLSP13_002227, MRBLSP13_002921 and MRBLSP13_003006). In the LSP13 background, we were able to generate a markerless triple deletion mutant of these homologs. We call the triple deletion mutant $\Delta 3bxdA$. Compared to the wildtype LSP13 strain, no colour change to red was observed by $\Delta 3bxdA$ in the assay with MBOA-containing plates. This indicated that $\Delta 3bxdA$ failed to form AMPO. We then confirmed the lack of AMPO formation by $\Delta 3bxdA$ in liquid cultures with complex and minimal media. Unlike wildtype, the mutant $\Delta 3bxdA$ failed to grow in minimal media with MBOA as a sole carbon source. These experiments reveal that BxdA is responsible for the degradation of MBOA and AMPO formation *in vivo*. Interestingly, we found that the triple mutant could still degrade MBOA in complex but not in the minimal medium. The mutant's degradation was slower

compared to the wildtype strain LSP13. This finding indicates that *Sphingobium* possesses another, BxdA-independent pathway to degrade MBOA that does not lead to the formation of AMPO. However, this pathway seems to be insufficient to allow growth on MBOA. This new data has been added as new Figures 5b-d and S11 to the manuscript and is described in the new paragraph starting in line 331.

2. The Bxd gene cluster encompasses 13 enzyme genes, yet most were challenging to characterize in vitro due to difficulties in protein expression in *E. coli*. Has the author explored other heterologous expression systems? Additionally, there is a lack of information on the substrate specificity of BxdA. Is it capable of metabolizing other lactones present in the rhizosphere?

With the identification of *bxdA* gene we have focussed for the moment only on this gene of the cluster. The other genes of the cluster and exploring other heterologous expression systems present ground for future work.

The reviewer asking whether BxdA is “capable of metabolizing other lactones present in the rhizosphere?” raises a very interesting question. For the moment, we do not know which lactones are present in the rhizosphere of maize or are prevalent among the abundant bacteria on maize roots. We have not performed a systematic biochemical characterization of the BxdA protein with other compounds. Indicative towards substrate specificity, we have tested the purified BxdA protein with BOA, the non-methoxylated relative of MBOA. Compared to the empty vector control, BOA concentrations were reduced by BxdA and low levels of APO were formed (APO is the non-methoxylated relative of AMPO). Please note that for this experiment, no APO standard was available and thus no quantification of APO formation was possible. However, APO formation was detected based on the calculated mass (APO 213.000; see extracted ion chromatogram) and the fragmentation spectrum was matching the database entry for APO. Compared to MBOA, BOA was less degraded during the same incubation time and using the same assay settings (same protein amounts and same initial concentrations of (M)BOA). This indicates that BxdA is less efficient in converting BOA to APO than converting MBOA to AMPO under the tested conditions. This new data has been added as new Figure S11c to the manuscript and is described in the Supplementary results.

3. Maize root secretes DIMBOA-Glu, which is converted to AMPO. The authors highlight the common occurrence of the deglycosylation step in soil bacteria, and certain root-colonizing members can use MBOA as a carbon source, resulting in the formation of AMPO. In soil, AMPO is further metabolized to produce 2-acetylamino-9-methoxy-2-amino-3H-phenoxazin-3-one (AAMPO). Does the maize rhizosphere harbor bacteria that metabolize AMPO? Additionally, are AMPO levels higher than AAMPO in the rhizosphere soil of maize? Do the rhizosphere soil of maize contain more AMPO than AAMPO? It would be beneficial for readers if the authors discussed their findings within the framework of metabolite profiles present in the maize rhizosphere soil.

Our early measurements of AMPO and AAMPO in the maize rhizosphere revealed low levels of AMPO (~0.1 µg/kg) while AAMPO levels were below the limit of detection (Hu et al. 2018). Even with an improved protocol (not published yet), quantification remains difficult with the detected low levels. Regarding the question, whether the maize root bacteria can further metabolize AMPO to AAMPO – yes, this is the case and therefore we have left the information in the manuscript. However, we think that including the topic of acetylation (maize bacteria metabolising AMPO to AAMPO) in the discussion distracts from the main focus of the work, which is AMPO formation.

Minor comments

1. Fig.3. Carbons released in the catabolism from MBOA to AMP would be considerably less compared to those from DIMBOA-Glu or glucose at the same concentration. I would recommend the authors to show the growth curve of the bacterial isolates in Supplementary Figures not just AUC for easier comparison of growth based on the different substrates.

As recommended, we show now the growth curves of the bacterial isolates for easier comparison of growth based on the different substrates (see new Fig. S6).

2. There are both AMPO-forming and non AMPO-forming members in the Family. This aspect should be explained in Fig S4 as well.

This is correct. We have added a reminder that mappings may contain AMPO-forming and non AMPO-forming strains to the corresponding paragraph in the manuscript text (Line 186).

3. Fig. S8E. What does “gene” in X axis mean?

The dot plot reports the location (y-axis) of the significantly upregulated genes (in response to MBOA) in the genome of LMB2. The genes are reported on the x-axis and are sorted following their position in the genome. The graph reports that many neighbouring genes around the base position of 3000 in the genome are upregulated in response to MBOA. This group of genes belong to the *bxl* gene cluster and are marked with a black box. We have changed the x-axis title to “Upregulated gene” (now Fig. S9e).

REVIEWER #3 (REMARKS TO THE AUTHOR):

In the current study, the authors aimed to address the mechanisms by which root-associated bacteria metabolize root-secreted benzoxiazinoids. Using the accumulation of the red color-producing metabolite, AMPO, the authors identified a series of bacterial strains that are able to accumulate AMPO when treated with MBOA, one of the abundant benzoxiazinoids secreted from maize roots. This capacity was enriched in maize-derived isolates compared to the isolated from other plant species and exerted a growth fitness in MBOA-catalyzing isolates, implying that this is an adaptive feature of maize-associated microbiota. Then, by employing integrative bioinformatic approaches, including comparative genomics and transcriptomics, they identified the bxd cluster that possibly mediates MBOA catabolism. Although its *in vivo* function remains to be seen, one of the gene products, bxdA, was shown to possess MBOA-catalyzing activity *in vitro*, supporting this hypothesis. Overall, this study revealed a molecular framework that might have allowed maize plants to recruit their specific structure of root microbiota. The manuscript is well-written and well-structured, and the details are paid careful attention. To what extent this system contributes to maize fitness, for example, by exerting fitness effect, etc, remains to be seen but is an intriguing open question. Although not explicitly raised as criticism, Reviewer#3 mentions that the *in vivo* function of BxdA remains to be seen. See response to Reviewer#2. In brief, we have generated a markerless triple deletion mutant of the three bxdA homologs in *Sphingobium* LSP13. The mutant $\Delta 3\text{bxdA}$ fails to form AMPO on MBOA-containing plates as well as in complex and minimal liquid media with MBOA. Collectively, these experiments reveal that BxdA – also from *Sphingobium* and not only *Microbacterium* – is responsible for AMPO formation *in vivo* and *in vitro*. Furthermore, the experiments revealed that the $\Delta 3\text{bxdA}$ mutant failed to grow on MBOA as sole carbon source.

An important question that remains to be addressed is how this capacity has been acquired in the studied *Microbacteria* lineage. It is almost certain that this gene cluster was horizontally acquired, and the authors also showed that there are some other bacterial strains that have BxdA homologs. The authors concluded that BxdA was rarely found in other bacteria, but I do not see this, as Figure S9B clearly identified genes with high (>60%) similarity with BxdA, which ranged from Actinobacteria to Beta/Gamma-proteobacteria. It was also unclear why the authors focused on the top 30 hits. Also, the comparison within *Microbacteria* lineage was limited to maize and *Arabidopsis* isolates, and many other strains were not taken into account. Overall, the analysis of BxdA gene distribution is limited to a biased genome dataset and does not support the authors' claim. To claim that it is a specific innovation in the MBOA-catabolizing isolates associated with the roots of benzoxazinoid-producing plants (lines 414-415), it is necessary to perform a more comprehensive and unbiased genomic search for BxdA using a custom set of publicly available genome sequences.

We agree with the reviewer that the initial analysis of BxdA gene distribution was limited in breath and that it did not sufficiently support our claim. A related criticism was also raised by Reviewer#1. We have, therefore, performed a comprehensive and unbiased genomic search for BxdA by blasting the amino acid sequence of BxdA from LMB2 against a large set of publicly available genomes. We chose to query the collection of the Integrated Microbial Genomes / Microbiome (IMG/M) database of the Joint Genome Institute, where genomes of 123'000 isolated bacteria are hosted (state December 2023). The BLAST search identified 462 hits of which only 28 had sequence similarities of >60 %. These hits were mostly from plant and soil environments, while the isolation origin of the majority of genomes was not annotated. The highly similar BxdA hits were found in few bacterial taxa including Microbacteriaceae, Sphingomonadaceae, Micrococcaceae, Burkholderiaceae and Pseudomonadaceae, most of which we had identified and confirmed to form AMPO with our

isolate collection approach. We had hoped that the metadata of the IMG/M genomes would allow a systematic analysis of compartment (soil vs. root vs...) and host origin (plant species). Unfortunately, the sparse metadata of the few hits precluded to reach such conclusion (See Dataset S5). The results of the new homology search are now reported in the new Figure S12c.

I am also curious to know how the authors put this finding into the microbial community context. Apparently, not all bacterial strains catabolize DIMBOA-Glc down to AMPO but only until MBOA, while some bacteria catabolize only MBOA but not DIMBOA-Glc. These findings, depicted in lines 207-213, clearly suggest metabolic complementation between root microbiota. In this context, I would be curious to know whether those only catabolizing DIMBOA-Glc and those only catabolizing MBOA would co-occur in natural communities, in which case the community-context metabolic complementation can be clearly suggested. Also, in this context, I was wondering if AMPO produced by these root bacteria can be further utilized by other bacteria and, alternatively, if AMPO is abundantly detected in maize rhizosphere without further catabolism.

This comment touches on a highly interesting observation. Yes indeed, the screening of metabolic capacities revealed that the different metabolic traits are divided up among individual microbiota members and this suggests that they could cooperate or complement each other to metabolize the maize exudates. Synthetic community (SynCom) experiments would be the suitable approach allowing experimentally to test for such metabolic complementation. For sake of a clear focus on AMPO formation in this manuscript we decided not to elaborate on this topic in this study.

Regarding the question whether AMPO produced by root bacteria can be further utilized by other bacteria – yes, this is the case. In brief, from liquid culture experiments, we know that some strains (not only AMPO-formers) have the capacity to acetylate AMPO to AAMPO. Please see the response to Reviewer#2 above for more details. Hence, we know that bacteria have the capacity for further metabolism e.g. to AAMPO. Furthermore, our time-course data indicates that bacteria like LMD1 have the capacity to also degrade AMPO.

Below are some minor comments:

- Something is wrong with the Fig 1 legend. Please revise it to fit with its content.

Done.

- How do the authors explain the difference in AMPO production of Arabidopsis isolates between plate- and liquid-based screening?

Plate and liquid assays differ in multiple aspects: the plate assay uses solid 100% tryptic soy broth agar (TSA) plates, was supplemented with 200 mg/mL (~1'200 μ M) of MBOA was scored after 10d of growth. In contrast, the liquid assay was scored within less than 3 days and consisted of 50% liquid tryptic soy broth (TSB) containing 500 μ M of MBOA. Hence the plate assay was allowing more time to metabolize MBOA; whereas AMPO formation was scored from lower starting amounts of MBOA in shorter time in the liquid assay. While such distinctions in the assays do not matter much for rapid AMPO formers like most maize strains are, they matter for slow or weaker MBOA-degraders like the Arabidopsis bacteria.

REVIEWERS' COMMENTS

Reviewer #1 (Remarks to the Author):

Thank the authors for the effect they have put into updating. After the authors' explanation of the comments and the revisions to the MS, particularly the adjustment of the Result, the coherence of the paper and the completeness of the narrative have been significantly improved.

There are still some minor issues remaining:

1. Statistical modifications have been applied to Figure 3d. The small black dots above the bars may indicate significant differences. It is recommended to include a legend for better reader comprehension.

2. Line 458-462: The authors observed the enrichment of maize root-specific microbiota in the rhizosphere due to their ability to utilize MBOA as a carbon source. The lactonase encoded by *bxdA* is pivotal in converting MBOA into AMPO.

Mention of Lactonase often prompts associations with chemical signaling between plants and microorganisms, as the authors also noted " Lactonases occur in various bacteria⁴⁵ and typically degrade N-acyl homoserine lactones, which are signalling metabolites of bacterial quorum sensing." Therefore, the reduction of QS may enhance the host's fitness. Here, I have a minor query: Is it plausible that MBOA or its metabolite, AMPO, serve as signaling molecules to stimulate dialogue between specific microorganisms and host plants, potentially enhancing host fitness? I didn't request additional experimental work; rather, I simply engaged in an open discussion with the authors on this topic.

3. Line 553, "1:10⁻¹ and 1:10⁻⁴", -1 and -4 require superscript.

4. Supplementary Line 44-45 and 47 The diagram marked in the text (eg. Fig. S4a and Fig. S4b) does not match the figure S4 below. When reading a good work and getting immersed in the story, encountering such errors in the text can be quite frustrating... It's like a wonderful concert is suddenly interrupted.

Reviewer #2 (Remarks to the Author):

I appreciate the authors for addressing the comments. In particular, the authors have generated the d3bxd mutant of *Sphingobium* and clearly demonstrated that bxd is involved in metabolic adaptation to the benzoxazinoid-rich environment. I have no additional comments to improve this manuscript, but I would suggest changing the title. The previous title may not be appropriate without *in vivo* characterization, but now the authors have performed additional experiments to reveal its *in vivo* function. I prefer the previous title. At least this second title is too specific. This manuscript is not focused on its biochemical characterization of benzoxazinoid catabolism, but also targeted to microbial ecology perspectives.

Reviewer #3 (Remarks to the Author):

In the revised version of this manuscript, the authors have addressed all my concerns. The genetic disruption of bxdA genes in *Sphingomonas* isolates have greatly improved the basis of this work. Yet, it should be noted that their *in planta* function remains to be addressed. It could have been great if the authors tested this by inoculating WT and bx1 plants with WT and bxdA mutant bacteria in both gnotobiotic and natural soil conditions. The authors proposed a few potential scenarios how this molecular process might provide benefit to the plant hosts, and I look forward to seeing them tested experimentally, as I understand that this will be the major focus of their next work.

Point-by-point response letter on revised manuscript

“The lactonase BxdA mediates metabolic specialisation of maize root bacteria to benzoxazinoids” by Thoenen et al.

We appreciate the positive perception of our revised work by the editor and the reviewers. We have addressed all major and minor concerns and incorporated them in the revised version #2 of our manuscript. Overall, we are grateful for the critical and rigorous assessment of our manuscript and we see that it has substantially improved with the reviewer’s constructive feedback. For transparency, we provide the revised manuscript in a ‘clean’ version and in a version that highlights the ‘changes’ compared to the original submission using track/change.

Here we respond to the reviewers’ comments on our revised manuscript in a detailed point-by-point manner. Our responses are coloured in blue. The line numbers mentioned refer to the re-revised main text document without tracked changes.

Reviewer #1 (Remarks to the Author):

Thank the authors for the effort they have put into updating. After the authors’ explanation of the comments and the revisions to the MS, particularly the adjustment of the Result, the coherence of the paper and the completeness of the narrative have been significantly improved.

There are still some minor issues remaining:

1. Statistical modifications have been applied to Figure 3d. The small black dots above the bars may indicate significant differences. It is recommended to include a legend for better reader comprehension.

We have increased the size of the asterisks in the figure and added the statistical information to the legend text.

2. Line 458-462: The authors observed the enrichment of maize root-specific microbiota in the rhizosphere due to their ability to utilize MBOA as a carbon source. The lactonase encoded by bxdA is pivotal in converting MBOA into AMPO. Mention of Lactonase often prompts associations with chemical signaling between plants and microorganisms, as the authors also noted "Lactonases occur in various bacteria⁴⁵ and typically degrade N-acyl homoserine lactones, which are signalling metabolites of bacterial quorum sensing." Therefore, the reduction of QS may enhance the host's fitness. Here, I have a minor query: Is it plausible that MBOA or its metabolite, AMPO, serve as signaling molecules to stimulate dialogue between specific microorganisms and host plants, potentially enhancing host fitness? I didn't request additional experimental work; rather, I simply engaged in an open discussion with the authors on this topic.

The reviewers points to a very interesting discussion. Since BxdA is annotated as a N-acyl homoserine lactonase family protein and because of its biochemistry (hydrolyses an ester bond of a lactone) it is indeed tempting to speculate that there is a link with quorum sensing. We see multiple possibilities for further research. (i) Plants may interfere with bacterial behaviour through BxdA, (ii) BxdA may have evolved from “QS-lactonases”s because of similar chemistry but functions today in MBOA degradation without effects on bacterial behaviour or (iii) it may also point to the possibility that benzoxazinoids would function as signalling molecules between bacteria. Regarding (i) there is evidence for a direct role of benzoxazinoids (and other plant specialised metabolites) inhibiting bacterial biofilm (e.g., doi:10.3390/molecules21101397, doi:10.1111/1462-2920.15216 or doi.org/10.1007/s00253-018-8787-x). We have started new research looking into a link between BxdA, benzoxazinoids and quorum sensing.

3. Line 553, “1:10⁻¹ and 1:10⁻⁴”, -1 and -4 require superscript.

Done.

4. Supplementary Line 44-45 and 47 The diagram marked in the text (eg. Fig. S4a and Fig. S4b) does not match the figure S4 below. When reading a good work and getting immersed in the story, encountering such errors in the text can be quite frustrating... It's like a wonderful concert is suddenly interrupted.

We agree with the reviewer that wonderful concert should not be suddenly interrupted. We corrected the figure indexing.

Reviewer #2 (Remarks to the Author):

I appreciate the authors for addressing the comments. In particular, the authors have generated the d3bxd mutant of *Sphingobium* and clearly demonstrated that bxd is involved in metabolic adaptation to the benzoxazinoid-rich environment. I have no additional comments to improve this manuscript, but I would suggest changing the title. The previous title may not be appropriate without in vivo characterization, but now the authors have performed additional experiments to reveal its in vivo function. I prefer the previous title. At least this second title is too specific. This manuscript is not focused on its biochemical characterization of benzoxazinoid catabolism, but also targeted to microbial ecology perspectives.

We implemented the reviewer's suggestion for changing the title. We agree that the study not only covers the biochemistry but also microbial ecology. Therefore, we replaced “metabolic adaptation” in the previous title and the new title now reads: “The lactonase BxdA mediates metabolic specialisation of maize root bacteria to benzoxazinoids”.

Reviewer #3 (Remarks to the Author):

In the revised version of this manuscript, the authors have addressed all my concerns. The genetic disruption of *bxdA* genes in *Sphingomonas* isolates have greatly improved the basis of this work. Yet, it should be noted that their *in planta* function remains to be addressed. It could have been great if the authors tested this by inoculating WT and bx1 plants with WT and *bxdA* mutant bacteria in both gnotobiotic and natural soil conditions. The authors proposed a few potential scenarios how this molecular process might provide benefit to the plant hosts, and I look forward to seeing them tested experimentally, as I understand that this will be the major focus of their next work.

We agree with the reviewer that the *in planta* function(s) remains to be tested and we have added this comment to the discussion of the manuscript (L491).